# Towards Adversarially Robust VLMs with an Information-Theoretic Approach

## Abstract

Vision–Language Models (VLMs) derive their zero-shot ability from tight alignment between image and text representations, which can be viewed through the lens of mutual information (MI). This alignment is fragile: VLMs are vulnerable both to subtle pixel-level adversarial attacks and to typographic attacks in which overlaid text hijacks predictions. Existing defenses are isolated solutions, relying on proxy objectives tailored to each threat. We hypothesize that both attack types can be viewed as degrading cross-modal mutual information (MI), and we propose an information-theoretic framework that explicitly mitigates this effect under adversarial perturbations. We first prove a bound that links adversarial risk to the *MI gap*, defined as the reduction in MI between clean and perturbed image–text views. Building on this, we derive a practical, differentiable objective that minimizes an upper bound on the MI gap using a neural MI estimator, yielding a single, attack-agnostic training scheme. Empirically, our method improves robustness to both pixel-space and typographic attacks within a single fine-tuning pipeline, outperforming prior methods while maintaining competitive accuracy on clean inputs. These results show that explicitly preserving cross-modal MI is a principled and effective path to robust VLMs.

## 1 Introduction

Recent advancements in Vision-Language foundation Models (VLMs) have demonstrated remarkable value across a spectrum of vision tasks, extending beyond academic benchmarks to real-world applications in manufacturing, autonomous driving, and defect detection (Minderer et al., 2022);(Zhou et al., 2024); (Li et al., 2024). This widespread adoption into safety-critical domains has, in turn, ignited concerns regarding their reliability and trustworthiness Vu & Lai (2025). The power of VLMs like CLIP (Radford et al., 2021a) stems from their ability to learn a shared embedding space where visual and textual concepts are aligned, a process akin to maximizing the mutual information (MI) between the two modalities, often via an InfoNCE objective (van den Oord et al., 2018).

In spite of the success in semantic alignment between modalities, this cross-modal alignment is often fragile in some hostile settings. VLMs are susceptible to two seemingly distinct classes of attacks that degrade their performance. The first is the well-documented threat of pixel-space adversarial attacks, where small, imperceptible perturbations to an image can cause misclassification (Goodfellow et al., 2015). The second is the typographic attack, where overlaying misleading text onto an image can hijack the model's prediction (Goh et al., 2021), exploiting its tendency to prioritize textual cues over visual evidence. These vulnerabilities pose significant risks, as a model's failure in an autonomous vehicle or on a factory inspection line can have severe consequences (National Transportation Safety Board, 2019).

Current defenses are typically developed and evaluated for a single threat model at a time. Adversarial training frameworks such as TRADES (Zhang et al., 2019) focus on smoothing decision boundaries against small pixel-level perturbations. VLM-specific adaptations, including FARE (Schlarmann et al., 2024) and TGA-ZSR (Yu et al., 2024), regularize feature embeddings or attention maps to improve robustness to pixel-wise attacks, while typographic attacks are usually handled by a separate line of work. Instead of designing yet another threat-specific objective, we take a different perspective: we argue that, regardless of whether the perturbation is global and low-variance (pixel-

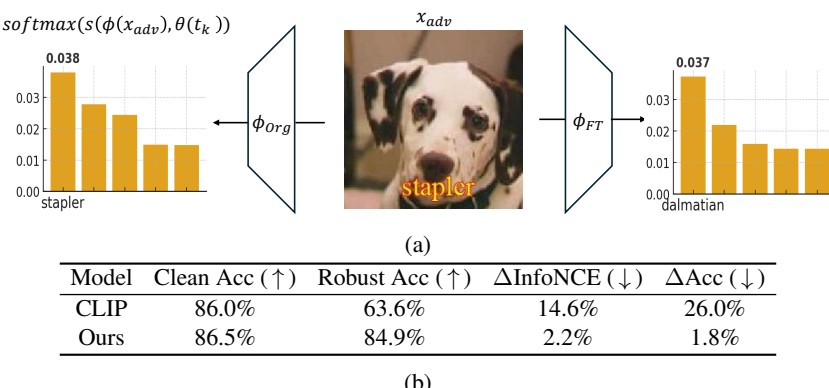

(a)

| Model | Clean Acc ($\uparrow$) | Robust Acc ($\uparrow$) | $\Delta$InfoNCE ($\downarrow$) | $\Delta$Acc ($\downarrow$) |
|-------|-----------|------------|-----------|---------|
| CLIP | 86.0% | 63.6% | 14.6% | 26.0% |
| Ours | 86.5% | 84.9% | 2.2% | 1.8% |

(b)

Figure 1: **(a)** Softmax output under a typographic overlay: writing the token "stapler" on the image ($x_{adv}$) steers CLIP's frozen encoder $\phi_{org}$ toward the wrong prototype, whereas our fine-tuned encoder $\phi_{FT}$ restores alignment and peaks at the correct class ("dalmatian"). **(b)** Evaluation results on ImageNet (Deng et al., 2009) samples: our method markedly improves robust accuracy while keeping the MI-gap proxy ($\Delta$InfoNCE) and accuracy drop ($\Delta$Acc) small.

level) or localized and high-variance (typographic), successful attacks tend to disrupt the alignment between visual and textual representations, which manifests as a reduction in image–text mutual information. This suggests a common cause underlying heterogeneous threats. We therefore derive an information-theoretic training objective that directly minimizes the mutual-information gap between clean and perturbed views, and we instantiate the *same* objective with both pixel and typographic perturbations, showing that explicitly preserving image–text mutual information yields favorable clean–robust trade-offs across these threat types.

This paper advocates that a more principled approach to VLM robustness is to directly safeguard cross-modal mutual information, complementing existing threat-specific robustness efforts. Both pixel-space and typographic attacks can be viewed through a single lens as operations that corrupt this shared information between modalities. Therefore, a more fundamental defense is to formulate a learning objective that explicitly preserves MI under worst-case perturbations whereas the previous researches counter specific attack algorithms case-by-case. By leveraging tools from information theory, such as the Donsker–Varadhan representation of KL-divergence that underpins neural MI estimators like MINE (Belghazi et al., 2018), we can construct a single objective that is applied across heterogeneous perturbation views, without designing separate loss functions for each threat model. This reframes adversarial defense in our setting as preserving cross-modal information under worst-case perturbations.

**Our Contributions.** Building upon this information-theoretic foundation, we introduce InfoGap, a novel framework for enhancing VLM robustness. Our primary contributions are threefold:

- We derive, from a risk-theoretic perspective, an upper bound that links adversarial decision-boundary risk to a mutual-information gap between clean and perturbed image–text views, motivating MI-based objectives for robust alignment.
- We propose an analytically tractable and practical upper bound for the *information gap* created by adversarial attacks. We further develop a method to compute this bound and incorporate it directly into the training process as a learnable objective, inspired by neural MI estimators.
- We empirically demonstrate that our single fine-tuning framework improves robustness to both pixel-space adversarial attacks and semantic typographic attacks, and is competitive with or better than specialized state-of-the-art methods on a range of benchmarks. These findings are consistent with the view that robust cross-modal alignment can benefit multiple threat models.

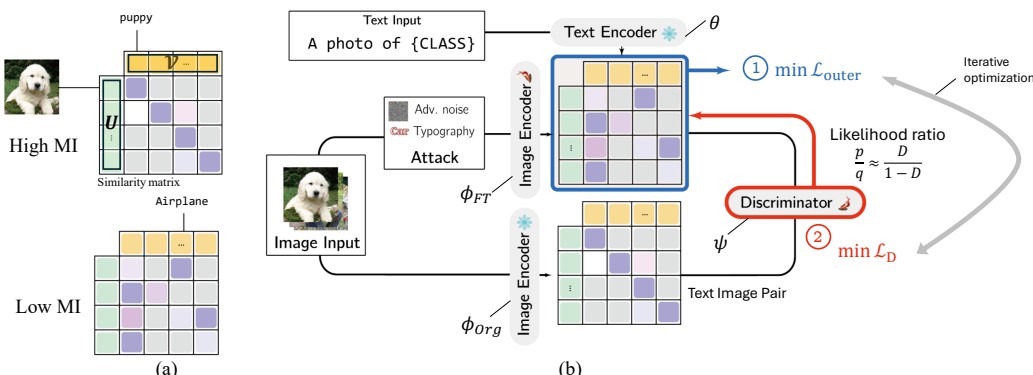

(a)         (b)

Figure 2: (a) Cross-modal MI: when the image embedding aligns with its correct text prototype (e.g., puppy), similarity concentrates on the positive column (high MI); with a mismatched label (e.g., airplane), alignment weakens and similarity spreads to non-targets (low MI). (b) InfoGap overview: a frozen text encoder provides prototypes; pixel or typographic perturbations generate a perturbed view; a trainable vision encoder and discriminator minimize the MI gap with feature anchoring to achieve robustness.

## 2 PRELIMINARIES AND PROBLEM SETUP

**Adversarial Attack.** We consider two threat models. (1) *Pixel-wise adversarial attacks* add small, human-imperceptible perturbations to the image, a setting popularized by Goodfellow et al. (2015) and commonly instantiated with Projected Gradient Descent (PGD) (Madry et al., 2018) under an $\ell_\infty$ budget $\varepsilon$. (2) *Typographic attacks* overlay misleading text onto the image, causing label flip even when the visual content is unchanged (Goh et al., 2021); see Fig. 1 for an illustration.

**Encoders and Text Prototypes.** Let $\theta(\cdot)$ denote the *frozen* CLIP text encoder. We precompute a *bank of text prototypes* for the $K$ classes: let $\mathcal{Y} = \{1, \ldots, K\}$ be the label set and, for each class $k \in \mathcal{Y}$, define a prompt string $t_k$ (e.g., "a photo of a {class name}") and its embedding

$$v_k := \theta(t_k) \in \mathbb{R}^d.$$

The collection $\mathcal{V} = \{v_k\}_{k=1}^K$ is what we call the *bank of text prototypes*; it is fixed throughout training. (If multiple prompt templates are used, $v_k$ denotes their mean embedding; results are unchanged if a single template is used.) On the vision side, $\phi_{\text{org}}$ is the pretrained (frozen) CLIP image encoder used as a stable anchor, and $\phi_{\text{FT}}$ is its trainable copy.

**Data and Views.** Given an image $X$ with label $L \in \mathcal{Y}$, define the clean image embedding $U := \phi_{\text{org}}(X) \in \mathbb{R}^d$ and the class prototype $V^+ := v_L \in \mathbb{R}^d$. Let $\mathcal{A}$ be a family of image perturbations (pixel-space or typographic). For $a \in \mathcal{A}$, write $X^{(a)} := a(X)$ and $U_{\text{adv}} := \phi_{\text{FT}}(X^{(a)})$. We define the clean and perturbed views as

$$Z := (U, \mathcal{V}) \quad \text{(clean)}, \qquad Z_{\text{adv}} := (U_{\text{adv}}, \mathcal{V}) \quad \text{(perturbed)}.$$

Let $p(u, v)$ and $q(u, v)$ denote the joint densities of $(U, V)$ under the clean and perturbed pipelines, where $V$ is the random variable for the text prototype corresponding to the image's true label, denoted $V^+$. Set $w(u, v) := p(u, v)/q(u, v)$. For any joint $f$ over $(U, V)$, $I_f(U, V)$ denotes the mutual information.

**Mutual Information Estimation via the Donsker–Varadhan (DV) Representation.** Computing $I(U, V)$ exactly is typically intractable. The DV representation expresses MI as a variational problem:

$$I(U, V) = D_{\text{KL}}(p(u, v) \| p(u)p(v)) = \sup_T \left\{ \mathbb{E}_{p(u,v)}[T(u, v)] - \log \mathbb{E}_{p(u)p(v)}[e^{T(u,v)}] \right\}.$$

This converts MI estimation into maximizing a sample-based objective over a function class for $T$. In practice, one approximates the supremum by restricting $T$ to a parametrized family and optimizing from samples (e.g., neural critics as in MINE (Belghazi et al., 2018)), yielding a differentiable estimator usable in downstream objectives.

**Natural vs. Adversarial risks.** Following Zhang et al. (2019), the robust risk decomposes as $R_{\mathrm{rob}}(f) = R_{\mathrm{nat}}(f) + R_{\mathrm{bdy}}(f)$, where $R_{\mathrm{bdy}}$ is the probability that a perturbation within the threat set drives a clean sample across the decision boundary (i.e., causes a label flip). Under adversarial evaluation, the primary objective is therefore to *reduce* $R_{\mathrm{bdy}}$: learn decision regions with larger margins and posteriors that remain stable under admissible perturbations, so that clean points stay well inside their class regions and do not cross boundaries. In the sequel we make this notion precise by relating $R_{\mathrm{bdy}}$ to distances between clean and perturbed posteriors (and associated information-theoretic quantities), yielding tractable surrogates for directly lowering flip probability under attack.

## 3 METHODOLOGY

### 3.1 DECISION-BOUNDARY RISK AND MUTUAL INFORMATION

**Claim.** We show that an upper bound on the gap between adversarial risk and natural risk can be expressed in terms of a *mutual information (MI) gap* between clean and perturbed image–text views. We make this precise by tying the MI gap to an upper bound on the decision–boundary (flip) risk.

**Generalization of boundary risk** Here, we generalize the boundary risk as the probability of a label flip caused by an adversary. Let $\widehat{L}(Z)$ denote the predicted label given view $Z$. Then, we define the boundary risk as $R_{\mathrm{bdy}} = \Pr(\widehat{L}(Z) = L, \widehat{L}(Z_{adv}) \neq L)$ where $R_{\mathrm{nat}} = \Pr(\widehat{L}(Z) \neq L)$ and $R_{\mathrm{adv}} = \Pr(\widehat{L}(Z_{adv}) \neq L)$. As detailed in Appendix A, this risk is upper-bounded by the expected Total Variation (TV) distance between the posterior distributions of the clean and adversarial views:

$$R_{\mathrm{bdy}} \leq \frac{2}{\tau_0} \mathbb{E}\big[\|p(L \mid Z) - p(L \mid Z_{adv})\|_{\mathrm{TV}}\big], \tag{1}$$

where $\tau_0$ is the minimum classification margin. Next, using Pinsker's inequality, the expected TV distance can itself be upper-bounded by Conditional Mutual Information (CMI) terms:

$$\mathbb{E}\big[\|p(L \mid Z) - p(L \mid Z_{adv})\|_{\mathrm{TV}}\big] \leq \sqrt{\frac{1}{2}I(L; Z_{adv} \mid Z)} + \sqrt{\frac{1}{2}I(L; Z \mid Z_{adv})}. \tag{2}$$

Combining these bounds directly yields our main theoretical result.

**Theorem 3.1** (General bound without label-agnosticity). *Under the margin assumption with parameter $\tau_0 > 0$, for any attack mechanism,*

$$R_{\mathrm{bdy}} \leq \frac{\sqrt{2}}{\tau_0}\left(\sqrt{I(L; Z_{adv} \mid Z)} + \sqrt{I(L; Z \mid Z_{adv})}\right).$$

*(Proof in Appendix A.)*

**Decomposing the Bound via the MI Gap.** Using the chain rule for mutual information, we can decompose the term $I(L; Z \mid Z_{adv})$ into the MI gap and the conditional MI term $\epsilon = I(L; Z_{adv} \mid Z)$. This leads to a more general corollary that explicitly frames the boundary risk in terms of the MI gap and this residual information term $\epsilon$. Here, as defined in Sec. 2, $V$ and $U$ are the embedded text and image vectors.

**Corollary 3.2** (Approximate label-agnostic bound). *The boundary risk is bounded by:*

$$R_{\mathrm{bdy}} \leq \frac{\sqrt{2}}{\tau_0}\left(\sqrt{\epsilon} + \sqrt{I(V^+; U) - I(V^+; U_{\mathrm{adv}}) + \epsilon}\right),$$

*where MI gap $= I(V^+; U) - I(V^+; U_{\mathrm{adv}})$, and $\epsilon = I(L; Z_{adv} \mid Z)$.*

*(Proof in Appendix A.)*

This result highlights that the boundary risk bound depends on both the MI gap and the residual term $\epsilon$, and reduces to a purely MI-gap–based form as $\epsilon \to 0$, matching the strict label-agnostic case.

*Remark* 3.3. The term $\epsilon = I(L; Z_{adv} \mid Z)$ quantifies how much information the adversarial view $Z_{adv}$ provides about the label $L$ beyond what the clean view $Z$ already provides. Since $I(L; Z_{adv} \mid$

$Z) \leq H(L \mid Z)$, the value of $\epsilon$ is upper-bounded by the model's uncertainty on clean data. For a pretrained CLIP model with a low clean error rate $p_e$, Fano's inequality implies that this uncertainty satisfies $H(L \mid Z) \leq h(p_e) + p_e \log(K - 1)$. This suggests that, when clean accuracy is high, the MI-gap term can play a substantial role in the bound, although this interpretation is heuristic and depends on how tight these inequalities are in practice.

## 3.2 MUTUAL INFORMATION GAP ESTIMATION

We now derive our main training objective by introducing a variational upper bound on the mutual-information gap between the clean and perturbed views,

$$\Delta_{\mathrm{MI}} := I(V^+; U) - I(V^+; U_{\mathrm{adv}}).$$

For brevity, we will drop the superscript and write $V \equiv V^+$ (i.e., the text prototype of the true label), so the gap will be denoted $I(V; U) - I(V; U_{\mathrm{adv}})$. Equivalently, letting $p$ and $q$ denote the clean and perturbed joints over $(U, V)$, the distribution-indexed form is $I_p(U, V) - I_q(U, V)$.

**Proposition 3.4** (Upper bound on the MI gap). *Let $p$ and $q$ denote the joint pdfs of $(U, V)$ under the clean and adversarial settings, respectively, and define the density ratio*

$$w(u, v) := \frac{p(u, v)}{q(u, v)}.$$

*If the text marginal is unaffected by attack, i.e., $p_V = q_V$, then*

$$I_p(U, V) \leq I_q^{\mathrm{IW}}(U, V) + D_{\mathrm{KL}}(p\|q), \tag{3}$$

*where the importance-weighted mutual information under $q$ is*

$$I_q^{\mathrm{IW}}(U, V) = \mathbb{E}_q\left[w(u, v) \log \frac{q(u, v)}{q(u)\, q(v)}\right].$$

*Consequently,*

$$I_p(U, V) - I_q(U, V) \leq \underbrace{I_q^{\mathrm{IW}}(U, V) - I_q(U, V)}_{\textit{information term}} + \underbrace{D_{\mathrm{KL}}(p\|q)}_{\textit{distribution term}}. \tag{4}$$

*(Proof in Appendix A.)*

Intuitively, the first term measures how much *information* is preserved under $q$ once we reweight by $w = p/q$; the second term quantifies the *distribution mismatch*. Jointly reducing both terms is expected to tighten the MI gap bound and to counteract robustness-induced alignment decay.

**Proposition 3.5** (Donsker-Varadhan Bound for Importance-Weighted MI). *Let $q$ denote the joint distribution of $(U, V)$ under the fine-tuned model, and let $w(u, v) = \frac{p(u,v)}{q(u,v)}$ be the density ratio between the clean and fine-tuned joints. The importance-weighted mutual information, $I_q^{\mathrm{IW}}$, admits the following variational lower bound based on the Donsker–Varadhan representation:*

$$I_q^{\mathrm{IW}} = \sup_{T:\mathcal{U}\times\mathcal{V}\to\mathbb{R}} \left(\mathbb{E}_q[w\, T] - \log\left(\mathbb{E}_{q(u)\,q(v)}[w\, e^T]\right)\right). \tag{5}$$

*(Proof in Appendix A.)*

**Critic Choice & Surrogate Gap.** Given the Donsker-Varadhan bound in Prop. 3.5, we consider two options for the critic $T$: (i) optimize $T$ (MINE-style) to tighten bounds, or (ii) reuse an inference similarity as a *fixed* critic for stability and cost of training. In either case, we measure the MI *gap* via

$$\Delta_{\mathrm{gap}} := I_q^{\mathrm{IW}} - I_q, \qquad \widehat{\Delta}_{\mathrm{gap}}(T) := J_q^{\mathrm{IW}}(T) - J_q(T),$$

where $J_q(T) = \mathbb{E}_q[T] - \log \mathbb{E}_{q(u)q(v)}[e^T]$ and $J_q^{\mathrm{IW}}(T) = \mathbb{E}_q[wT] - \log \mathbb{E}_{q(u)q(v)}[we^T]$. We adopt (ii) and justify it by showing that the fixed-critic surrogate error $\widehat{\Delta}_{\mathrm{gap}}(T) - \Delta_{\mathrm{gap}}$ admits a simple bound (Observation 3.6 ) that vanishes as $w \to 1$.

**Observation 3.6** (Bias bound for a fixed-critic MI-gap surrogate). *Let $w(u, v) = \frac{p(u,v)}{q(u,v)}$ and $|T| \leq M$. Define $J_q(T) = \mathbb{E}_q[T] - \log \mathbb{E}_{q(u)q(v)}[e^T]$ and $J_q^{\mathrm{IW}}(T) = \mathbb{E}_q[wT] - \log \mathbb{E}_{q(u)q(v)}[we^T]$. Then*

$$\left(J_q^{\mathrm{IW}}(T) - J_q(T)\right) - \left(I_q^{\mathrm{IW}} - I_q\right) \leq 2\left(M\, \mathbb{E}_q|w - 1| + e^{2M}\, \mathbb{E}_{q(u)q(v)}|w - 1|\right).$$

The derivation is given in Appendix A.5.

**NCE-based Ratio Estimation.** We estimate the density ratio $w(u, v) = p(u, v)/q(u, v)$ via noise-contrastive estimation with a binary discriminator. Let $r \in \{1, 0\}$ indicate whether $(u, v)$ comes from $p$ (clean) or $q$ (perturbed), with equal priors. By Bayes' rule, the optimal discriminator is $D^\star(u, v) = \Pr(r = 1 \mid u, v) = p/(p + q)$, hence $w(u, v) = \frac{p(u,v)}{q(u,v)} = \frac{D^\star(u,v)}{1 - D^\star(u,v)}$. In practice, we parameterize the discriminator $D_\psi$ as a simple multi-layer perceptron (MLP) and train it with a binary cross-entropy (BCE) loss to distinguish between clean and perturbed embedding pairs. We then use its output $\widehat{w}(u, v) = D_\psi(u, v)/(1 - D_\psi(u, v))$ inside the importance-weighted MI estimator and the divergence term. Details regarding implementations are deferred to Appendix A.6

**Stable Surrogate for the KL Divergence.** A standard likelihood-ratio estimator for $D_{\mathrm{KL}}(p\|q)$, which relies on minimizing $\mathbb{E}_q[w \log w]$, is notoriously unstable for training the density ratio. Its gradient, $1 + \log w$, vanishes at $w = e^{-1}$, failing to provide a robust anchoring force towards the desired equilibrium at $w = 1$. We therefore optimize a more stable surrogate based on the $\chi^2$-divergence, whose gradient is proportional to $(w - 1)$ and provides a direct pull towards $w = 1$. Observation 3.7 and Figure 6 provide theoretical and empirical justification for this choice, and its influence on the total loss is gradually increased during training via a simple annealing schedule for its weight $\lambda$.

**Observation 3.7** (Anchoring via $\chi^2$). *Let $p, q$ be joint densities on $(u, v)$ with $q > 0$ a.e., and set $w(u, v) = p(u, v)/q(u, v)$. Then $\mathbb{E}_q[w] = \iint w \, q \, du \, dv = \iint p \, du \, dv = 1$. The chi-square divergence is $\chi^2(p\|q) = \iint \frac{(p-q)^2}{q} \, du \, dv = \mathbb{E}_q[(w - 1)^2]$. Using $\log u \le u - 1$ for $u > 0$ gives $\mathbb{E}_q[w \log w] \le \mathbb{E}_q[w(w-1)] = \mathbb{E}_q[(w-1)^2] = \chi^2(p\|q)$. Thus $\chi^2$ upper-bounds the KL divergence term with likelihood ratio and, with gradient $\propto (w - 1)$, pulls $w$ toward $1$ for stable training.*

**Preventing Concept Drift** Recent evidence shows that conventional fine-tuning can substantially erode the broad, transferable knowledge embedded in foundation models—a phenomenon termed *concept forgetting* (Mukhoti et al., 2024). In particular, Mukhoti et al. (2024) demonstrate that end-to-end adaptation often degrades recognition of concepts outside the downstream task, and that explicitly preserving pre-trained features mitigates this collapse. Our setting exhibits the same risk: robustness-oriented updates can drift the vision encoder away from CLIP's zero-shot semantic structure. To counteract this drift, we anchor the fine-tuned image features to their pre-trained counterparts via an $\ell_2$ feature-preservation term, thereby retaining zero-shot capability while improving robustness.

### 3.3 Training Objective and Procedure

**Perturbed View.** Given an image $x$ and label $L$, let the clean anchor be $u := \phi_{\mathrm{org}}(x)$ and, for an attack $a \in \mathcal{A}$, define the perturbed embedding $u_{\mathrm{adv}} := \phi_{\mathrm{FT}}(x^{(a)})$. For pixel-wise attacks we use a label-agnostic inner objective $x^{(a)} \in \arg\max_{x' \in \mathcal{B}(x)} \left\| \phi_{\mathrm{FT}}(x') - u \right\|_2^2$, where $\mathcal{B}(x)$ is the admissible perturbation set (e.g., an $\ell_\infty$ ball of radius $\varepsilon$); this perturbs only the *visual* pathway and keeps the text side fixed, encouraging $I(L; Z_{\mathrm{adv}} \mid Z) \approx 0$ so that the boundary risk is governed solely by the MI gap. For typographic views, on the other hand, we construct $x^{(a)}$ by overlaying a mismatched label token on $x$ (with $y' \ne L$), leaving the text prototype bank unchanged; we then set $Z = (u, \mathcal{V})$ and $Z_{\mathrm{adv}} = (u', \mathcal{V})$.

**Training Objective and Procedure.** Let $v^+ = \theta(t_L)$ and let the density–ratio $w(u, v) = \frac{p(u,v)}{q(u,v)}$ be estimated by a discriminator. With a fixed similarity critic $T$, the encoder minimizes

$$\mathcal{L}_{\mathrm{outer}} = \left( \widehat{J}_q^{\mathrm{IW}}(T) - \widehat{J}_q(T) \right) + \lambda \widehat{\chi}^2(p\|q) + \gamma \frac{1}{B} \sum_{i=1}^{B} \left\| u_{\mathrm{adv}, i} - u_i \right\|_2^2, \tag{6}$$

where $\widehat{J}_q$, $\widehat{J}_q^{\mathrm{IW}}$, and $\widehat{\chi}^2$ are the minibatch estimators defined in the *Batch estimators* paragraph below. Optimization alternates two updates: the encoder minimizes $\mathcal{L}_{\mathrm{outer}}$, and the density–ratio discriminator minimizes a standard BCE loss on clean vs. perturbed pairs. This yields a GAN–style dynamic (Goodfellow et al., 2014): at equilibrium the discriminator approaches $50\%$ accuracy ($w \to 1$), indicating matched joints, while the encoder closes the MI gap without eroding zero-shot behavior.

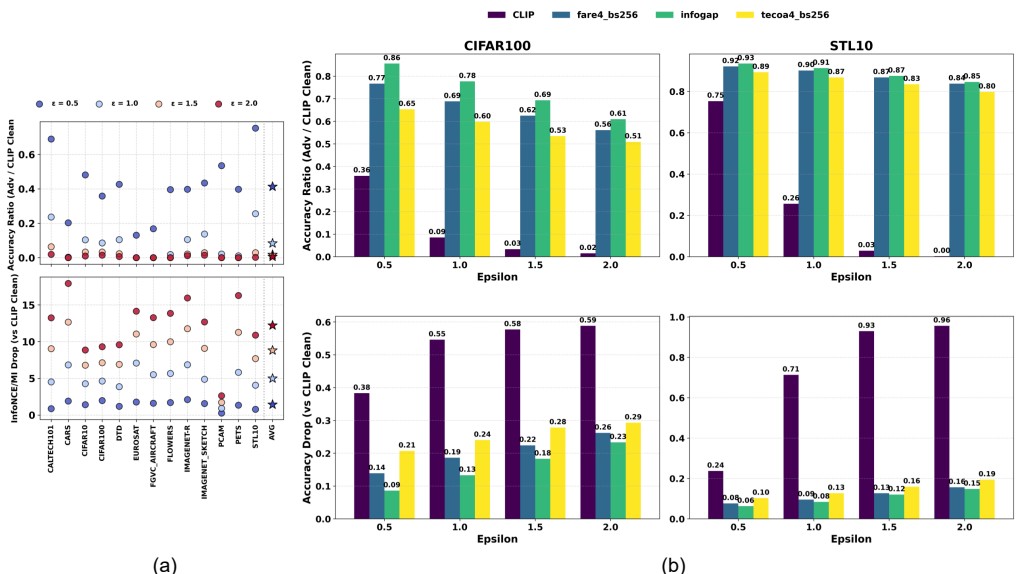

(a)                                                                    (b)

Figure 3: Empirical link between mutual-information drop and robust accuracy under pixel-space attacks. (a) For each dataset and $\epsilon \in \{0.5, 1.0, 1.5, 2.0\}/255$, we plot CLIP's accuracy ratio (adv/clean) and the corresponding InfoNCE drop. (b) On CIFAR-100 and STL-10, we compare CLIP, FARE, TeCoA, and InfoGap; InfoGap provides the best robustness–accuracy trade-off across perturbation strengths.

## 3.4 EMPIRICAL JUSTIFICATION FOR OUR APPROACH

We apply pixel-wise attacks of varying strength, $\epsilon \in \{0.5, 1.0, 1.5, 2.0\}/255$, and measure for each dataset the adversarial-to-clean accuracy ratio together with the drop in our MI estimator. As shown in Figure 3(a), larger perturbations simultaneously increase the MI drop and decrease the accuracy ratio, revealing a tight inverse correlation between the two. Figure 3(b) further shows that InfoGap mitigates this degradation more effectively than CLIP and remains competitive with specialized defenses (FARE, TeCoA) on CIFAR-100 and STL-10, supporting the view that minimizing the MI-gap is a principled way to preserve robust accuracy.

## 4 EXPERIMENTS

### 4.1 MODEL AND METRIC

We evaluate zero-shot classification with a CLIP ViT-B/32 encoder fine-tuned via a mutual-information objective for robustness to pixel-wise adversarial attack and typographic perturbations.

**Training.** For *pixel-wise* adversarial attacks, we use two budgets: $\epsilon = 2/255$ with batch size 128 and learning rate $1 \times 10^{-5}$, and $\epsilon = 4/255$ with batch size 256 and learning rate $3 \times 10^{-5}$. Unless otherwise noted, the remaining hyperparameters follow FARE; full details are provided in the Appendix. For typographic-robustness baselines based on preference optimization (PO), we adhere to Afzali et al. (Afzali et al., 2025) and train on ImageNet-100, a subset of ImageNet, with a batch size of 512 and a learning rate of $2 \times 10^{-5}$; further details are deferred to the Appendix A.6.

**Datasets.** Adversarial robustness is measured on ImageNet (Deng et al., 2009) and 13 zero-shot datasets used widely for CLIP evaluation: CIFAR-10/100 (Krizhevsky, 2009), STL-10 (Coates et al., 2011), Stanford Cars (Krause et al., 2013), Caltech-101 (Li et al., 2022), Oxford-IIIT Pets (Parkhi et al., 2012), Flowers-102 (Nilsback & Zisserman, 2008), DTD (Cimpoi et al., 2014), EuroSAT (Helber et al., 2019), FGVC Aircraft (Maji et al., 2013), PCam (Veeling et al., 2018), ImageNet-R (Hendrycks et al., 2021), and ImageNet-Sketch (Wang et al., 2019). Typographic robustness is evaluated on Caltech-101, Oxford-IIIT Pets, Stanford Cars, Flowers-102, FGVC Aircraft, DTD, EuroSAT, and SUN397 (Xiao et al., 2010).

Table 1: Zeroshot-clean (C) and Robust (R) accuracies (%) under a 100-step PGD attack with $\epsilon = 2/255$. *Zeroshot Average* is the mean across the evaluated zero-shot datasets. *Sum* denotes the trade-off score $C + R$ (reported on the R rows). Best value of each dataset are highlighted in **blue** for C and **red** for R.

| Method | Accuracy | CIFAR10 | STL10 | CIFAR100 | Cars | Caltech101 | Pets | Flowers | DTD | EuroSAT | FGVC | PCam | ImageNet-R | Sketch | Zeroshot Average | C+R |
|---|---|---|---|---|---|---|---|---|---|---|---|---|---|---|---|---|
| CLIP(Radford et al., 2021b) | C | 87.6 | 95.8 | 59.7 | 53.7 | 80.2 | 83.7 | 58.4 | 41.7 | 44.3 | 16.6 | 55.3 | 65.9 | 40.7 | 60.3 | |
| | R | 0.0 | 0.0 | 0.0 | 0.0 | 0.0 | 0.0 | 0.0 | 0.0 | 0.0 | 0.0 | 0.0 | 0.0 | 0.0 | 0.0 | 60.3 |
| TeCoA(Mao et al., 2022) | C | 73.9 | 87.6 | 42.2 | 9.6 | 68.5 | 72.2 | 19.7 | 22.4 | 17.3 | 5.6 | 50.1 | 45.8 | 29.9 | 41.9 | |
| | R | 56.3 | 76.4 | 30.4 | 5.8 | 63.3 | 58.1 | 13.0 | 16.3 | 8.2 | 3.4 | 50.0 | 33.7 | 21.4 | 33.6 | 75.5 |
| TGA-ZSR(Yu et al., 2024) | C | 75.0 | 89.7 | 43.1 | 10.6 | 70.6 | 73.4 | 20.9 | 23.7 | **20.1** | 6.1 | **50.2** | 45.7 | 30.2 | 43.0 | |
| | R | 56.8 | 78.5 | 31.3 | 6.4 | 64.2 | **60.4** | 13.5 | 18.3 | 10.8 | 3.2 | 50.1 | **34.2** | **21.9** | 34.6 | 77.6 |
| FARE(Schlarmann et al., 2024) | C | 73.3 | 90.0 | 49.6 | 34.5 | 78.7 | 78.0 | 31.4 | 30.8 | 15.9 | **10.0** | **50.2** | 48.6 | 31.7 | 47.9 | |
| | R | 54.5 | 80.2 | 33.6 | **15.0** | **69.6** | 55.0 | **18.0** | 22.5 | **13.5** | 5.4 | **50.2** | 32.2 | 21.1 | **36.2** | 84.1 |
| InfoGap (Ours) | C | **84.5** | **92.4** | **56.0** | 34.8 | **78.9** | 78.5 | **36.2** | **35.0** | 17.4 | 8.8 | 49.9 | **51.7** | **32.4** | **50.5** | |
| | R | **62.3** | **80.8** | **37.5** | 10.2 | 65.1 | 49.0 | 17.0 | **23.3** | 13.4 | 2.8 | 46.9 | 28.8 | 19.3 | 35.1 | **85.6** |

**Metrics.** We report zero-shot *top-1* accuracy using the standard CLIP template-prompt evaluation. *Clean* accuracy uses unperturbed images; *robust* accuracy applies test-time adversaries under the stated threat model. Unless noted, we use 50-step AutoAttack (Croce & Hein, 2020) in $\ell_\infty$ with $\epsilon \in \{2, 4\}/255$; 100 steps of PGD (Madry et al., 2018) and CW (Carlini & Wagner, 2016) results appear where indicated. In Table 2 and Table 3, $AA^2$ and $PGD^4$ denotes pixel-wise AutoAttack with $\epsilon$=2/255 and 100-step of PGD attack with $\epsilon$=4/255, respectively. All methods share the same prompt set, preprocessing, and zero-shot protocol.

Table 2: Zero-shot Average Accuracy (C+R, %) at training $\epsilon$=2/255. Each entry is Clean + Robust for the specified threat. Best and second-best are **bold** and underlined, respectively.

| Method | $AA^2$ | $AA^4$ | $PGD^2$ | $PGD^4$ | CW | Avg |
|---|---|---|---|---|---|---|
| TeCoA | 77.1 | 64.8 | 76.1 | 68.8 | 83.7 | 74.1 |
| FARE | 83.3 | 63.1 | **90.5** | **72.3** | **92.6** | 80.4 |
| TGA-ZSR | 82.4 | **68.3** | 82.3 | 71.4 | 85.0 | 77.9 |
| InfoGap (**Ours**) | **85.0** | 67.9 | 87.2 | 72.1 | 91.5 | **80.7** |

Table 3: Average Accuracy (C+R, %) for models trained with $\epsilon$=4/255.

| Method | $AA^2$ | $AA^4$ | $PGD^2$ | $PGD^4$ | CW | Avg |
|---|---|---|---|---|---|---|
| TeCoA | 74.4 | 65.2 | 75.5 | **77.7** | 77.6 | 74.1 |
| FARE | 82.5 | **71.2** | 78.1 | 73.3 | 87.1 | 78.4 |
| TGA-ZSR | 76.5 | 67.5 | 77.6 | 69.3 | 80.0 | 74.2 |
| InfoGap (**Ours**) | **83.9** | 69.4 | **85.6** | 71.8 | **88.6** | **79.9** |

## 4.2 RESULTS

### 4.2.1 PIXEL-WISE ADVERSARIAL ATTACK

Achieving adversarial robustness typically entails some loss of clean accuracy, since pushing the decision boundary away from perturbed inputs can distort the representation of unperturbed ones(Zhang et al., 2019). In zero-shot adversarial robustness, the key objective is therefore to maximize robustness gains while minimizing the degradation of clean capability. Across datasets, our fine-tuned image encoder generally achieves a favorable robustness–accuracy trade-off compared to the baselines, often yielding larger robustness improvements for a similar level of clean accuracy (Table 1, Table 4).

**Strength Under Unseen Attacks.** Although our model is fine-tuned using PGD, it generalizes well to stronger and structurally different attacks, including AutoAttack (AA) and the Carlini–Wagner (CW) attack. Moreover, while training is performed at perturbation radius $\epsilon = 2/255$ (Table 2), the model maintains a competitive robustness–accuracy trade-off even when evaluated at a larger radius $\epsilon = 4/255$ (Table 3), indicating robustness beyond the training threat model.

**Representation Analysis.** UMAP (McInnes et al., 2018) projections (Fig. 4) show that baseline models undergo large clean→adversarial shifts with noticeable inter-class mixing, indicating that perturbations push samples across decision regions. In contrast, *InfoGap* keeps adversarial embeddings close to their clean counterparts and preserves well-separated class clusters, suggesting that our objective maintains cross-modal information and prevents representation collapse under attack.

Table 4: Zero-shot benchmark on typographic attack (O: Original, T: Typographic). Best and second-best *excluding CLIP* are marked as **bold** and underlined, respectively.

| Method | Caltech101 O | T | OxfordPets O | T | StanfordCars O | T | Flowers102 O | T | FGVCAircraft O | T | DTD O | T | SUN397 O | T | EuroSAT O | T | Avg O | T |
|---|---|---|---|---|---|---|---|---|---|---|---|---|---|---|---|---|---|---|
| CLIP | 88.6 | 64.0 | 87.4 | 59.0 | 58.7 | 21.0 | 66.3 | 31.3 | 19.0 | 10.8 | 44.6 | 25.5 | 61.7 | 34.0 | 43.0 | 4.9 | 58.7 | 31.3 |
| Malerzynska+ | 80.5 | 74.7 | 75.0 | 63.6 | 40.3 | 15.8 | 51.9 | 35.0 | 13.2 | 8.3 | 36.3 | 33.0 | 51.1 | 39.5 | 37.3 | 16.2 | 48.3 | 35.8 |
| PAINT | 88.5 | 83.6 | 85.2 | 76.5 | 55.3 | 33.4 | 64.7 | 54.9 | 17.7 | 14.5 | **42.61** | 36.6 | 61.7 | 53.6 | 38.2 | 17.3 | 56.7 | 46.3 |
| Defense-Prefix | **89.28** | 79.5 | **87.22** | 72.9 | 57.47 | 28.6 | 63.8 | 44.1 | **19.26** | 14.5 | 40.6 | 31.6 | 61.4 | 43.5 | 43.9 | 9.9 | **57.87** | 40.6 |
| DPO | 87.5 | 85.4 | 85.3 | 79.7 | 56.0 | 34.3 | 56.6 | 55.7 | 16.2 | 13.9 | 39.4 | 38.5 | 61.0 | 56.3 | **49.33** | 28.3 | 56.4 | 49.0 |
| IPO | 85.7 | 83.8 | 85.3 | 80.4 | 53.7 | 35.0 | 54.5 | 52.8 | 18.0 | 15.9 | 40.5 | 39.9 | 61.91 | 58.1 | 46.1 | **43.23** | 55.7 | 51.1 |
| KTO | 87.7 | 86.0 | 85.4 | 81.0 | **57.76** | 37.0 | 59.1 | 58.0 | 17.3 | 15.6 | 40.7 | 40.33 | **62.52** | **59.01** | 46.26 | **36.94** | 57.1 | 51.7 |
| FARE | 88.9 | 87.38 | 86.12 | **83.09** | 57.1 | 38.89 | **66.07** | 63.10 | 18.08 | 16.20 | 39.4 | 38.2 | 57.9 | 56.0 | 39.8 | 35.6 | 56.7 | 52.31 |
| InfoGap (Ours) | **89.36** | **88.04** | 84.1 | 81.83 | 56.7 | **40.87** | 65.28 | **64.68** | 17.5 | **17.11** | **42.24** | **41.06** | 59.4 | 58.44 | 42.9 | 33.3 | 57.18 | **52.59** |

● Clean sample    ✕ Adversarial sample

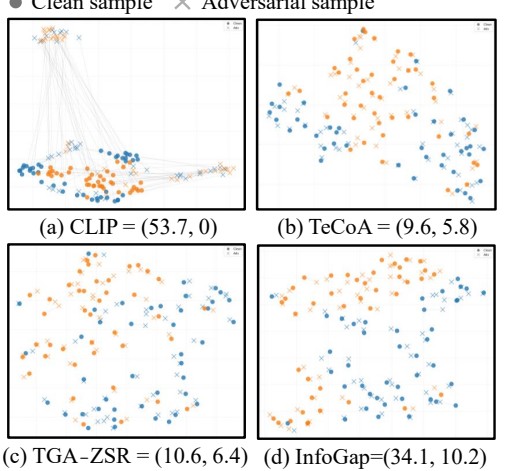

(a) CLIP = (53.7, 0)    (b) TeCoA = (9.6, 5.8)

(c) TGA–ZSR = (10.6, 6.4)    (d) InfoGap=(34.1, 10.2)

Figure 4: Visualization of baselines vs. *InfoGap* embeddings under pixel-wise adversarial attack. Each point is labeled by the predicted class and whether it is clean or adversarial.

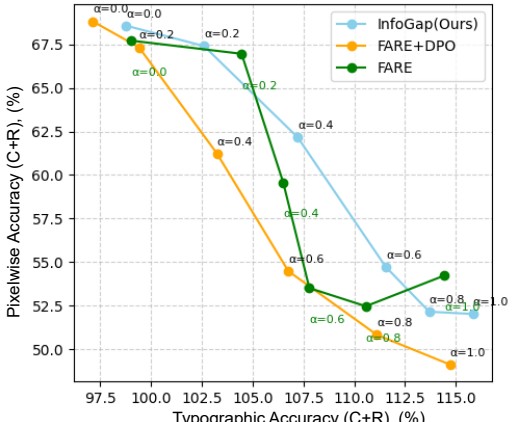

Figure 5: Trade-off of clean plus robust accuracy (C+R) for typographic vs. pixel-wise robustness as a function of $\alpha$ for *InfoGap* and the baseline checkpoint mixture.

### 4.2.2 TYPOGRAPHIC ATTACK

Table 4 summarizes zero-shot performance under typographic overlays. Baseline results are taken from prior work (Afzali et al., 2025) Across typography-focused baselines, InfoGap attains the highest typographic (T) accuracy on six of eight datasets and the best T average (52.59%). Its Original (O) average (57.18%) is within 1 point of the best clean method (Azuma & Matsui, 2023). On Caltech101, Flowers102, and DTD, **InfoGap** also achieves the best O accuracy, indicating that typography robustness does not sacrifice clean recognition. We further evaluate the pixel-wise defense FARE, trained with typographically perturbed images, on the same benchmark; in this cross-threat setting, InfoGap matches or slightly exceeds FARE in both O and T accuracy, suggesting that preserving cross-modal mutual information transfers more effectively across threat models than pixel-only objectives.

### 4.2.3 CLIP CHECKPOINT MIXING FOR A UNIFIED DEFENSE

We interpolate in weight space between a pixel-robust and a typo-robust checkpoint to obtain a single model that handles both threats. Following weight-space ensembling (Wortsman et al., 2022), we define the mixed checkpoint $\phi_{\text{ens}}(\alpha) = \alpha\,\phi_{\text{pixel}} + (1 - \alpha)\,\phi_{\text{typo}}$ and evaluate the robustness trade-off as a function of $\alpha$. For InfoGap, $\phi_{\text{pixel}}$ and $\phi_{\text{typo}}$ are checkpoints trained against pixel-wise and typographic attacks, respectively. As unified baselines, we consider mixtures where $\phi_{\text{pixel}}$ is the strongest pixel-wise FARE model and $\phi_{\text{typo}}$ is either (i) the strongest typographic model (DPO) or (ii) a FARE checkpoint fine-tuned on typographic overlays, so that $\phi_{\text{ens}}^{\text{baseline}}(\alpha) = \alpha\,\phi_{\text{pixel}}^{\text{FARE}} + (1 - \alpha)\,\phi_{\text{typo}}^{(\text{DPO or FARE})}$; all pixel-wise checkpoints are trained with $\epsilon = \frac{4}{255}$, matching the main setup. We report, per dataset, the sum of clean and robust accuracy ("C+R"): pixel-wise robustness is measured with AutoAttack at $\epsilon = \frac{2}{255}$, and typographic robustness follows the standard overlay protocol, with results averaged over seven zero-shot datasets (Fig. 5). Across $\alpha \in \{0.0, 0.2, 0.4, 0.6, 0.8, 1.0\}$, the

InfoGap mixture consistently dominates both baseline mixtures, achieving a better trade-off between typographic and pixel-wise robustness while preserving clean accuracy.

### 4.3 ABLATION STUDY

**Loss Components.**
We ablate five variants: **(i)** Information only; **(ii)** Information + KL (distribution term in Eq. 4); **(iii)** Information + $\chi^2$; **(iv)** Information + KL + $\ell_2$; **(v) full**: our final design choice. All are trained/evaluated under pixel-space attacks. In Fig. 6, KL-based settings (a,b) yield diverging discriminator scores (clean vs. adv), indicating persistent mismatch between two distributions, whereas $\chi^2 + \ell_2$ (c) ideally collapses to $\approx$ 0.5, i.e., $w \to 1$ and it suggests the two views become harder to distinguish on average. This matches our theory: replacing the likelihood-ratio penalty with $\chi^2(p\|q)$ supplies an anchoring gradient signal proportional to $(w-1)$, and the $\ell_2$ feature-preservation term curbs concept drift. Consistently, Table 5 shows monotonic gains under the same information term, and adding $\ell_2$ gives the best clean/robust trade-off.

Table 5: Ablation on loss components. Accuracy reported as averages across eval sets (%). (a) (c) is corresponding to the each entries in 6

| **Components** | Clean | AA$^2$ |
|---|---|---|
| Info. term | 30.83 | 12.71 |
| Info. term + KL - (a) | 33.36 | 23.88 |
| Info. term + $\chi^2$ | 34.36 | 24.06 |
| Info. term + KL + $\ell_2$ reg - (b) | 42.85 | 31.38 |
| Info. term + $\chi^2$ + $\ell_2$ reg - (c) | **45.78** | **32.24** |

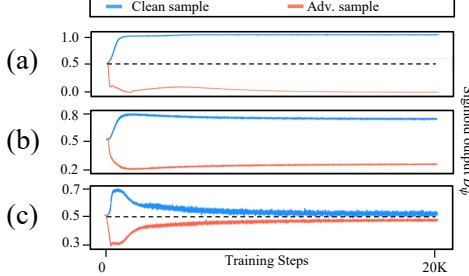

Figure 6: Median discriminator score over training for pixel-wise adversarial attack .

**Cost analysis.** All robustness methods considered share the same core per-step budget: one clean forward pass through the frozen image encoder, a $k$-step pixel-wise inner attack, and one forward/backward pass on adversarial images. Table 6 reports only the *additional* overhead on top of this shared budget. For DPO and KTO, the extra ViT pass comes from a frozen reference CLIP policy, while all ImageNet-1k text embeddings are precomputed and cached. For ViT-B/32 ($\approx$ 88M parameters, text bank size $C$=1000, embedding dimension $d$=512, patches $P$=49), InfoGap requires no extra ViT passes and adds only $\sim$ 0.79M trainable parameters ($<$ 1%), so its computational overhead is negligible.

| Method | Extra ViT passes per step | Extra compute (non-ViT) | Extra trainable params |
|---|---|---|---|
| **FARE** | +0 | L2 on embeddings $O(Bd)$ | None |
| **TeCoA** | +0 | logits vs. text bank $O(BdC)$ | None |
| **TGA-ZSR** | +3 | patch–text dot product $O(BP+Bd)$ | None |
| **InfoGap (ours)** | +0 | discriminator & MI on $(B{\times}d)$ | $\approx$**0.79M** |
| **DPO** | +1 (frozen ref CLIP) | log-ratio losses $O(B)$ | None |
| **KTO** | +1 (frozen ref CLIP) | sigmoid losses $O(B)$ | None |
| **PPO** | +0 (cached old logits) | ratio/clipping $O(B)$ | None |

Table 6: Additional per-step overhead beyond the shared training budget: extra ViT passes, non-ViT compute, and trainable parameters.

## 5 CONCLUSION

We present an information-theoretic framework that aims to address both pixel-space and typographic attacks in a common formulation. By directly minimizing an adversarial mutual-information gap, our method often matches or surpasses baselines(FARE, TeCoA) while preserving clean accuracy. Our results support the view that explicitly safeguarding cross-modal information is a principled and practically useful way to improve robustness in multimodal settings.

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

You may include other additional sections here.

# A    PROOFS FOR SECTION 3.1

## A.1    SUPPORTING LEMMA

**Lemma A.1** (Lower bound on TV under label flip via the margin). *Let $p, q \in \Delta^{K-1}$. Write $i^\star = \arg\max_k p_k$ and $j^\star = \arg\max_k q_k$. If $i^\star \neq j^\star$, then*

$$\|p - q\|_{\mathrm{TV}} \geq \frac{p_{i^\star} - \max_{k \neq i^\star} p_k}{2}.$$

*Proof.* Set $\Delta_k := p_k - q_k$. Then

$$\|p - q\|_{\mathrm{TV}} = \tfrac{1}{2} \sum_k |\Delta_k| \geq \tfrac{1}{2}\big(|\Delta_{i^\star}| + |\Delta_{j^\star}|\big) \geq \tfrac{1}{2}\big((p_{i^\star} - q_{i^\star}) + (q_{j^\star} - p_{j^\star})\big),$$

where the last step uses $|x| \geq x$ and $|x| \geq -x$. Since $j^\star = \arg\max q_k$ and $i^\star = \arg\max p_k$,

$$q_{j^\star} \geq q_{i^\star} \quad \text{and} \quad p_{i^\star} \geq p_{j^\star} \quad \Rightarrow \quad p_{j^\star} \leq \max_{k \neq i^\star} p_k.$$

Hence

$$(p_{i^\star} - q_{i^\star}) + (q_{j^\star} - p_{j^\star}) \geq (p_{i^\star} - p_{j^\star}) + (q_{j^\star} - q_{i^\star}) \geq p_{i^\star} - p_{j^\star} \geq p_{i^\star} - \max_{k \neq i^\star} p_k.$$

Combining the displays gives the claim. $\square$

**Theorem 3.1** (General bound without label-agnosticity). *Under the margin assumption with parameter $\tau_0 > 0$, for any attack mechanism,*

$$R_{\mathrm{bdy}} \leq \frac{\sqrt{2}}{\tau_0} \left( \sqrt{I(L; Z_{adv} \mid Z)} + \sqrt{I(L; Z \mid Z_{adv})} \right).$$

*Proof.* Let $p(L \mid Z)$ and $p(L \mid Z_{\mathrm{adv}})$ denote the posteriors for the clean and adversarial views, and write $\widehat{L}(Z) = \arg\max_k p_k(Z)$, $\widehat{L}(Z_{\mathrm{adv}}) = \arg\max_k p_k(Z_{\mathrm{adv}})$. Define the *clean posterior margin*

$$\gamma(Z) := p_{\widehat{L}(Z)}(Z) - \max_{k \neq \widehat{L}(Z)} p_k(Z),$$

and assume it is bounded away from zero: there exists $\tau_0 > 0$ such that $\gamma(Z) \geq \tau_0$ almost surely.

Next, consider the flip event

$$A := \{(Z, Z_{adv}, L) : \widehat{L}(Z) = L, \ \widehat{L}(Z_{adv}) \neq L\}.$$

By Lemma A.1, whenever $\widehat{L}(Z) \neq \widehat{L}(Z_{adv})$ we have

$$\big\|p(L \mid Z) - p(L \mid Z_{adv})\big\|_{\mathrm{TV}} \geq \frac{\gamma(Z)}{2}.$$

Since $A \subseteq \{\widehat{L}(Z) \neq \widehat{L}(Z_{adv})\}$, this yields the event inclusion

$$A \subseteq \left\{ (L, Z, Z_{adv}) : \|p(L \mid Z) - p(L \mid Z_{adv})\|_{\mathrm{TV}} \geq \tfrac{\gamma(Z)}{2} \right\}. \tag{7}$$

Using the bound $\mathbf{1}_{\{X \geq a\}} \leq X/a$ for all $X \geq 0$ and $a > 0$, and applying it to the event inclusion in Eq. (8) with $X = \|p(L \mid Z) - p(L \mid Z_{\text{adv}})\|_{\text{TV}}$ and $a = \gamma(Z)/2$, we obtain the pointwise inequality

$$\mathbf{1}_A \leq \tfrac{2}{\gamma(Z)} \big\|p(L \mid Z) - p(L \mid Z_{\text{adv}})\big\|_{\text{TV}}. \tag{$*$}$$

Taking expectations in equation $*$ yields

$$R_{\text{bdy}} = \mathbb{E}\big[\mathbf{1}_A\big] \leq \frac{2}{\tau_0}\,\mathbb{E}\big[\big\|p(L \mid Z) - p(L \mid Z_{adv})\big\|_{\text{TV}}\big]. \tag{8}$$

To convert the TV term into conditional mutual information, we use the triangle inequality followed by Pinsker's inequality:

$$\|p(L \mid Z) - p(L \mid Z_{adv})\|_{\text{TV}} \leq \|p(L \mid Z) - p(L \mid Z, Z_{adv})\|_{\text{TV}} + \|p(L \mid Z, Z_{adv}) - p(L \mid Z_{adv})\|_{\text{TV}}$$

$$\leq \sqrt{\tfrac{1}{2}\,\text{KL}\big(P_{L|Z,Z_{adv}} \,\|\, P_{L|Z}\big)} \;+\; \sqrt{\tfrac{1}{2}\,\text{KL}\big(P_{L|Z,Z_{adv}} \,\|\, P_{L|Z_{adv}}\big)}.$$

Taking expectations of the above and using Jensen's inequality for the concave function $\sqrt{\cdot}$ (i.e., $\mathbb{E}[\sqrt{X}] \leq \sqrt{\mathbb{E}[X]}$) leads to the bound in equation 2:

$$\mathbb{E}[\|p(L \mid Z) - p(L \mid Z_{adv})\|_{\text{TV}}] \leq \sqrt{\tfrac{1}{2}\,\mathbb{E}_{(L,Z,Z_{adv})}\Big[D_{\text{KL}}(P_{L|Z,Z_{adv}}\|P_{L|Z})\Big]}$$

$$+ \sqrt{\tfrac{1}{2}\,\mathbb{E}_{(L,Z,Z_{adv})}\Big[D_{\text{KL}}(P_{L|Z,Z_{adv}}\|P_{L|Z_{adv}})\Big]}$$

$$= \sqrt{\tfrac{1}{2}\,\mathbb{E}_{(Z,Z_{adv})}\Big[D_{\text{KL}}(P_{L|Z,Z_{adv}}\|P_{L|Z})\Big]}$$

$$+ \sqrt{\tfrac{1}{2}\,\mathbb{E}_{(Z,Z_{adv})}\Big[D_{\text{KL}}(P_{L|Z,Z_{adv}}\|P_{L|Z_{adv}})\Big]}$$

(since each KL term is a function of $(Z, Z_{adv})$ only)

$$= \sqrt{\tfrac{1}{2}\,I(L;Z_{adv} \mid Z)} \;+\; \sqrt{\tfrac{1}{2}\,I(L;Z \mid Z_{adv})}.$$

$$\square$$

*Remark* A.2 (CMI as an expectation of conditional KL). By definition,

$$I(L;Z_{\text{adv}} \mid Z) = \mathbb{E}_Z\Big[D_{\text{KL}}\big(P_{L,Z_{\text{adv}}|Z} \,\|\, P_{L|Z}\,P_{Z_{\text{adv}}|Z}\big)\Big]$$

$$= \mathbb{E}_Z\Big[\mathbb{E}_{Z_{\text{adv}}|Z}\big[D_{\text{KL}}\big(P_{L|Z,Z_{\text{adv}}} \,\|\, P_{L|Z}\big)\big]\Big]$$

$$= \mathbb{E}_{Z,Z_{\text{adv}}}\Big[D_{\text{KL}}\big(P_{L|Z,Z_{\text{adv}}} \,\|\, P_{L|Z}\big)\Big].$$

Likewise,

$$I(L;Z \mid Z_{\text{adv}}) = \mathbb{E}_{Z,Z_{\text{adv}}}\Big[D_{\text{KL}}\big(P_{L|Z,Z_{\text{adv}}} \,\|\, P_{L|Z_{\text{adv}}}\big)\Big].$$

Hence, each KL term that appears inside the square roots in the bound is a function of $(Z, Z_{\text{adv}})$ only, matching the definition of conditional mutual information.

## A.2 PROOF OF COROLLARY 3.2

**Corollary 3.2** (Approximate label-agnostic bound). *The boundary risk is bounded by:*

$$R_{\text{bdy}} \leq \frac{\sqrt{2}}{\tau_0}\left(\sqrt{\epsilon} + \sqrt{I(V^+;U) - I(V^+;U_{\text{adv}}) + \epsilon}\right),$$

*where MI gap = $I(V^+;U) - I(V^+;U_{\text{adv}})$, and $\epsilon = I(L;Z_{adv} \mid Z)$.*

*Proof.* We start from the general bound in Theorem 3.1. Our goal is to express the term $I(L; Z \mid Z_{adv})$ in terms of the MI gap. We use the chain rule for mutual information:

$$I(L; Z, Z_{adv}) = I(L; Z) + I(L; Z_{adv} \mid Z).$$
$$I(L; Z, Z_{adv}) = I(L; Z_{adv}) + I(L; Z \mid Z_{adv}).$$

Equating these two gives:

$$I(L; Z) + I(L; Z_{adv} \mid Z) = I(L; Z_{adv}) + I(L; Z \mid Z_{adv}).$$

Rearranging for $I(L; Z \mid Z_{adv})$ yields:

$$I(L; Z \mid Z_{adv}) = I(L; Z) - I(L; Z_{adv}) + I(L; Z_{adv} \mid Z).$$

Now, we specialize the general MI gap, $I(L; Z) - I(L; Z_{adv})$, to our CLIP zero-shot setting. Since the text prototype bank $\mathcal{V}$ is fixed, it provides no information about the label $L$ when the image embedding $U$ is known, thus $I(L; Z) = I(L; U)$. Furthermore, the label $L$ is in a one-to-one correspondence with its text prototype $V^+ = v_L$, so by MI invariance to bijections, $I(L; U) = I(V^+; U)$. Applying these transformations, we get:

$$I(L; Z) - I(L; Z_{adv}) = I(V^+; U) - I(V^+; U').$$

Let us define MI gap, $\Delta_{\mathrm{MI}} := I(V^+; U) - I(V^+; U_{\mathrm{adv}})$ and the residual CMI term $\epsilon := I(L; Z_{adv} \mid Z)$. Substituting these into the expression for $I(L; Z \mid Z_{adv})$ gives:

$$I(L; Z \mid Z_{adv}) = \Delta_{\mathrm{MI}} + \epsilon.$$

Finally, we substitute this and the definition of $\epsilon$ back into the original bound from Theorem 3.1:

$$R_{\mathrm{bdy}} \leq \frac{\sqrt{2}}{\tau_0} \left( \sqrt{I(L; Z_{adv} \mid Z)} + \sqrt{I(L; Z \mid Z_{adv})} \right)$$
$$= \frac{\sqrt{2}}{\tau_0} \left( \sqrt{\epsilon} + \sqrt{\Delta_{\mathrm{MI}} + \epsilon} \right).$$

This completes the proof. $\qquad\square$

## A.3 PROOF OF PROPOSITION 3.4

**Proposition 3.4** (Upper bound on the MI gap). *Let $p$ and $q$ denote the joint pdfs of $(U, V)$ under the clean and adversarial settings, respectively, and define the density ratio*

$$w(u, v) := \frac{p(u, v)}{q(u, v)}.$$

*If the text marginal is unaffected by attack, i.e., $p_V = q_V$, then*

$$I_p(U, V) \leq I_q^{\mathrm{IW}}(U, V) + D_{\mathrm{KL}}(p \| q), \tag{3}$$

*where the importance-weighted mutual information under $q$ is*

$$I_q^{\mathrm{IW}}(U, V) = \mathbb{E}_q \left[ w(u, v) \log \frac{q(u, v)}{q(u) q(v)} \right].$$

*Consequently,*

$$I_p(U, V) - I_q(U, V) \leq \underbrace{I_q^{\mathrm{IW}}(U, V) - I_q(U, V)}_{\text{information term}} + \underbrace{D_{\mathrm{KL}}(p \| q)}_{\text{distribution term}}. \tag{4}$$

*Proof.* Assume $p_V = q_V$. Write $w(u, v) = \frac{p(u,v)}{q(u,v)}$. Then

$$I_p(u, v) = \mathbb{E}_p \left[ \log \frac{p(u, v)}{p_U(U) p_V(V)} \right]$$
$$= \mathbb{E}_q \left[ w \log \frac{q(u, v)}{q_U(U) q_V(V)} \right] + \mathbb{E}_p \left[ \log \frac{p(u, v)}{q(u, v)} \right] + \mathbb{E}_p \left[ \log \frac{q_U(U) q_V(V)}{p_U(U) p_V(V)} \right]$$
$$= I_q^{\mathrm{IW}}(u, v) + D_{\mathrm{KL}}(p \| q) - D_{\mathrm{KL}}(p_U \| q_U),$$

where we used $p_V = q_V$ so $\mathbb{E}_p[\log(q_V / p_V)] = 0$. Since $D_{\mathrm{KL}}(p_U \| q_U) \geq 0$,

$$I_p(u, v) \leq I_q^{\mathrm{IW}}(u, v) + D_{\mathrm{KL}}(p \| q).$$

Subtract $I_q(u, v) = \mathbb{E}_q \left[ \log \frac{q(u,v)}{q_Z(Z) q_V(V)} \right]$ from both sides to obtain the MI-gap bound. $\qquad\square$

## A.4 PROOF OF PROPOSITION 3.5

**Proposition 3.5** (Donsker-Varadhan Bound for Importance-Weighted MI). *Let $q$ denote the joint distribution of $(U, V)$ under the fine-tuned model, and let $w(u, v) = \frac{p(u,v)}{q(u,v)}$ be the density ratio between the clean and fine-tuned joints. The importance-weighted mutual information, $I_q^{\mathrm{IW}}$, admits the following variational lower bound based on the Donsker–Varadhan representation:*

$$I_q^{\mathrm{IW}} = \sup_{T:\mathcal{U}\times\mathcal{V}\to\mathbb{R}} \left( \mathbb{E}_q\big[w\,T\big] \; - \; \log\Big(\mathbb{E}_{q(u)\,q(v)}\big[w\,e^T\big]\Big) \right). \tag{5}$$

*Proof.* By definition and using $p_Y = q_Y$ (no text attack),

$$I_q^{\mathrm{IW}}(u, v) := \mathbb{E}_q\left[ w(u, v) \log \frac{q(u, v)}{q(u)\,q(V)} \right].$$

Introduce a positive normalizer $Z_\star$ and write

$$I_q^{\mathrm{IW}} = \underbrace{\mathrm{KL}\Big( q(U, V)\,w \;\Big\|\; q(U)\,q(V)\,w\,Z_\star \Big)}_{\geq 0} + \log Z_\star.$$

Applying the Donsker–Varadhan (DV) variational form of KL,

$$\mathrm{KL}(P\|Q) = \sup_T \left\{ \mathbb{E}_P[T] - \log \mathbb{E}_Q\big[e^T\big] \right\},$$

with $P = q\,w$ and $Q = q(U)\,q(V)\,w\,Z_\star$, yields

$$I_q^{\mathrm{IW}} = \sup_T \left\{ \mathbb{E}_q[w\,T] - \log\big(Z_\star \cdot \mathbb{E}_{q(u)\,q(v)}[w\,e^T]\big) \right\} + \log Z_\star.$$

The $\log Z_\star$ terms cancel, giving the weighted DV objective $\mathbb{E}_q[w\,T] - \log \mathbb{E}_{q(u)\,q(v)}[w\,e^T]$. Fixing $T$ to the chosen similarity critic yields the practical estimator defined in A.6 after replacing the log-partition by its EMA.

$$\square$$

## A.5 ADDITIONAL DERIVATIONS

*Derivation of Observation 3.6.* (i) Since $|T| \leq M$, $|\mathbb{E}_q[wT] - \mathbb{E}_q[T]| \leq M\,\mathbb{E}_q|w - 1|$.
(ii) Let $A = \mathbb{E}_{q(u)q(v)}[we^T]$, $B = \mathbb{E}_{q(u)q(v)}[e^T]$. Because $e^T \in [e^{-M}, e^M]$, $|A - B| \leq e^M \mathbb{E}_{q(u)q(v)}|w - 1|$ and $\min\{A, B\} \geq e^{-M}$. By mean value theorem for log, $|\log A - \log B| \leq |A - B|/\min\{A, B\} \leq e^{2M}\mathbb{E}_{q(u)q(v)}|w - 1|$. Summing (i) and (ii) gives the claim. $\square$

## A.6 TRAINING IMPLEMENTATION

**Batch estimators.** For a minibatch $\{(u_i, v_i)\}_{i=1}^B$ and ratio $w_{ij} = \sigma(D_\psi(u_i, v_j))/(1 - \sigma(D_\psi(u_i, v_j)))$, the DV objectives and divergence surrogate are

$$\widehat{J}_q(T) = \tfrac{1}{B} \sum_i T(u_i, v_i) - \log\Big( \tfrac{1}{B^2} \sum_{i,j} e^{T(u_i, v_j)} \Big),$$

$$\widehat{J}_q^{\mathrm{IW}}(T) = \tfrac{1}{B} \sum_i w_{ii} T(u_i, v_i) - \log\Big( \tfrac{1}{B^2} \sum_{i,j} w_{ij} e^{T(u_i, v_j)} \Big),$$

$$\widehat{\chi}^2(p\|q) = \tfrac{1}{B} \sum_i \tfrac{(w_{ii}-1)^2}{w_{ii}}, \qquad \widehat{\mathcal{R}}_{\mathrm{feat}} = \tfrac{1}{B} \sum_i \|u_i' - u_i\|_2^2.$$

and $\mathcal{L}_{\mathrm{outer}} = \big(\widehat{J}_q^{\mathrm{IW}} - \widehat{J}_q\big) + \lambda\,\widehat{\chi}^2 + \gamma\,\widehat{\mathcal{R}}_{\mathrm{feat}}$. We use a fixed critic $T$ (scaled dot–product) for stability as mentioned in 3.2.

---

**Algorithm 1** InfoGap Training Step

---

**Require:** Vision encoder $\phi_{\text{FT}}$, frozen $\phi_{\text{org}}$, text encoder $\theta$, discriminator $D_\psi$, attack type
$\quad$ `attack_type` $\in \{\text{adv}, \text{typo}\}$, Gradient Penalty parameter $\lambda_{\text{GP}}$

1: Sample a mini-batch of images and labels $(x, y)$
2: $u_{\text{clean}}^{(\text{org})} \leftarrow \phi_{\text{org}}(x)$
3: **if** `attack_type` $=$ adv **then**
4: $\quad\quad x_{\text{adv}} \leftarrow \arg \max\limits_{\|x'-x\|_\infty \leq \varepsilon} \left\| \phi_{\text{FT}}(x') - \phi_{\text{Org}}(x) \right\|_2^2$ $\qquad\qquad\quad$ ▷ PGD inner maximization
5: **else**
6: $\quad\quad$ pick $y' \neq y$ and set $x_{\text{adv}} \leftarrow \text{OVERLAY}(x, t_{y'})$ $\qquad\qquad\quad$ ▷ Typographic view
7: **end if**
8: $u_{\text{adv}} \leftarrow \phi_{\text{FT}}(x_{\text{adv}}), \quad v \leftarrow \theta(t_y)$
9: $\widehat{w} \leftarrow \dfrac{D_\psi(u_{\text{adv}}, v)}{1 - D_\psi(u_{\text{adv}}, v)}$
10: Compute $I_q^{\text{IW}} - I_q$ and $\chi^2(p\|q)$ using $\widehat{w}$
11: **Update** $\phi_{\text{FT}}$ (freeze $D_\psi$):

$$\nabla_{\phi_{\text{FT}}}\Big([I_q^{\text{IW}} - I_q] + \lambda \chi^2(p\|q) + \gamma \|u_{\text{adv}} - u_{\text{clean}}^{(\text{org})}\|_2^2 \Big)$$

12: **Update** $D_\psi$ (freeze $\phi_{\text{FT}}$):

$$\nabla_{D_\psi}\Big(\mathcal{L}_{\text{BCE}}(D_\psi(z_{\text{clean}}^{(\text{org})}, v), 1) + \mathcal{L}_{\text{BCE}}(D_\psi(z_{\text{adv}}, v), 0) + \lambda_{\text{GP}}\, \mathcal{L}_{\text{GP}}\Big)$$

---

**MI Estimation via EMA.** The log-partition functions in the DV objectives (e.g., $\log(\mathbb{E}_{q(u)q(v)}[e^T])$) are intractable to compute over the full distribution. Following standard practice in neural MI estimation (Belghazi et al., 2018), we approximate them using an **exponential moving average (EMA)** of the batch-wise estimates, with a decay rate of $\alpha = 0.01$.

**Discriminator.** The MLP discriminator consists of three linear layers (Input—$d_x + d_y \to 512 \to 512 \to 1$) with Leaky ReLU activations (negative slope 0.05) and a dropout rate of 0.1 applied after the first two layers. It is trained using the Adam optimizer with parameters $\beta_1 = 0.5$ and $\beta_2 = 0.999$. To enhance training stability, we apply **spectral normalization** (Miyato et al., 2018) to all linear layers. We also implement a **gradient penalty** (Gulrajani et al., 2017) for stabilizing discriminator training with perturbation budget $\epsilon = \frac{2}{255}$.

**Loss Hyperparameters.** The final InfoGap objective in Eq. 6 uses the $\chi^2$ divergence for the distribution term. Its weight $\lambda$ is **linearly annealed from 0 to a final value of 10** over the training steps. The feature preservation term is weighted by a fixed $\gamma$, set to 5 for pixel-wise attacks and 20 for typographic attacks.

**Training details (pixel-space attacks).** All models are fine-tuned on ImageNet-1k starting from `openai` CLIP ViT-B/32 using AdamW with a cosine schedule, weight decay $1 \times 10^{-4}$, a total of 10,000 steps, and 700 warmup steps. Each iteration processes every image twice (clean and adversarial views). The only differences across budgets are batch size and learning rate:

$$\epsilon = \tfrac{2}{255} : \text{batch } 128, \text{ lr } 1 \times 10^{-5} \qquad \epsilon = \tfrac{4}{255} : \text{batch } 256, \text{ lr } 3 \times 10^{-5}.$$

Unless otherwise specified, inner maximization follows the FARE recipe: CLIP preprocessing, $\ell_\infty$ PGD with 10 steps, step size $1/255$, and a random start. We parameterize $\epsilon$ in pixel units and normalize by 255 in implementation.

**Training Objectives for Pixel-wise Robustness.** For each method, we outline the loss functions for inner maximization (attack generation) and outer minimization (model update).

**FARE** Adopts a pure feature-matching strategy where both the inner and outer objectives minimize the squared $L_2$ distance, $\mathcal{L} = \|u_{\text{adv}} - u\|_2^2$.

**TeCoA** Relies on a consistent classification objective, where both inner and outer loops minimize cross-entropy ($\mathcal{L}_{\mathrm{CE}}$) against text prototypes.

**TGA-ZSR** Extends the cross-entropy objective by enforcing alignment of *text-guided attention* maps between clean and adversarial views.

- **Pre-pooling features:** Let $\phi_g(x)$ denote the patch-level image features before the final pooling operation.
- **Text-guided attention:** The attention map is $A(x, t_y) := \mathrm{norm}(\phi_g(x) \cdot \theta(t_y)^\top)$.
- **Base Objective:** The standard cross-entropy loss on adversarial examples, $\mathcal{L}_{\mathrm{CE}} := \mathcal{L}_{\mathrm{CE}}(\mathrm{sim}(u_{\mathrm{adv}}, \mathcal{V})/\tau, y)$, where $u_{\mathrm{adv}} = \phi_{\mathrm{FT}}(x_{\mathrm{adv}})$.
- **Adversarial Alignment ($\mathcal{L}_{\mathbf{AR}}$):** $L_2$ distance between the attention map of the adversarial image (from the fine-tuned encoder) and that of the clean image (from the original encoder).

$$\mathcal{L}_{\mathrm{AR}} = \big\| A(\phi_{g,\mathrm{FT}}(x_{\mathrm{adv}}), t_y) - A(\phi_{g,\mathrm{org}}(x), t_y) \big\|_2$$

- **Clean Consistency ($\mathcal{L}_{\mathbf{AMC}}$):** $L_2$ distance between the attention maps of the clean image from both the fine-tuned and original encoders.

$$\mathcal{L}_{\mathrm{AMC}} = \big\| A(\phi_{g,\mathrm{FT}}(x), t_y) - A(\phi_{g,\mathrm{org}}(x), t_y) \big\|_2$$

- **Total Loss:** $\mathcal{L}_{\mathrm{TGA\text{-}ZSR}} = \mathcal{L}_{\mathrm{CE}} + \alpha\, \mathcal{L}_{\mathrm{AR}} + \beta\, \mathcal{L}_{\mathrm{AMC}}$, with hyperparameters $\alpha = 0.08$, $\beta = 0.05$ as per the original paper (Yu et al., 2024)

*Default* Here, D is $L_2$ distance.

**InfoGap (Ours)** Employs an information-theoretic objective with a learned discriminator.

- **Inner Loss (Attack):** $L_2$ distance, $\mathcal{L} = \|u_{\mathrm{adv}} - u\|_2^2$
- **Outer Loss (Training):** An information term regularized by $\chi^2$ divergence, $\mathcal{I}(\cdot) + \lambda\,\chi^2(p\|q)$, with linear annealing on $\lambda$.
- **Regularization:** A feature preservation term $\gamma\|u_{\mathrm{adv}} - z^{\mathrm{orig}}\|_2^2$ where $\gamma = 5$ in our reported version.

**Training details (typographic attacks).** For typographic robustness, we fine-tune the CLIP ViT-B/32 vision encoder on **ImageNet-100** for **3 epochs** with a batch size of **512**. We use the **AdamW** optimizer for both the vision encoder and the discriminator, applying a weight decay of $1 \times 10^{-4}$ to both. The vision encoder is trained with a learning rate of $1 \times 10^{-6}$, and the discriminator with $1 \times 10^{-5}$. Both learning rates follow a schedule with a **linear warmup** for the first 10

Table 7: Shared schedule for typographic adversarial training.

| Dataset | Batch Size / Epochs | Optimizer (LR) | Weight Decay | Loss Hyperparams |
|---|---|---|---|---|
| ImageNet-100 | 512 / 3 | AdamW (Encoder: 1e-6, Disc: 1e-5) | $1 \times 10^{-4}$ | $\lambda = 1.0, \gamma = 20$ |

# B BACKGROUND AND MOTIVATION

**Pretrained VLMs and the role of cross-modal information.** CLIP learns joint vision–language representations through contrastive alignment of images and texts. During training, each image–caption pair is treated as a positive match, while mismatched pairs serve as negatives. The training objective, a.k.a. a contrastive InfoNCE loss, pulls matching image and text embeddings together and pushes non-matching pairs apart. This process encourages a shared feature space where each modality's representation is highly predictive of the other, effectively maximizing the information the image and text share(Tschannen et al., 2020). In fact, minimizing the InfoNCE loss is equivalent to maximizing a lower bound on the mutual information (MI) between image and text representations(van den Oord et al., 2018; Poole et al., 2019). A lower InfoNCE loss implies a higher MI between the two modalities. Intuitively, CLIP's impressive zero-shot learning capability stems from this high cross-modal MI: the model learns rich visual features that retain semantic alignment with language descriptions. By maximizing MI across vision and language, CLIP embeds images and texts into a common space where true image–caption pairs stay close, enabling the model to recognize new classes from text descriptions alone.

**Why adversarial robustness is hard for VLMs.** CLIP is highly sensitive to small, imperceptible image perturbations that break image–text alignment and sharply reduce zero-shot accuracy (Schlarmann & Hein, 2023). Adversarial noise shifts the image embedding away from its correct textual neighbor, often via text-guided attention drift, causing misclassification (Yu et al., 2024). Recent defenses confirm this mechanism: contrastive, text-guided adversarial adaptation (TeCoA) and unsupervised adversarial fine-tuning of the CLIP vision encoder both improve zero-shot robustness by re-aligning perturbed images with their textual counterparts (Mao et al., 2022).

**Limitations of prior robustness approaches.** Prior robustness methods generally fall into two categories: defenses against pixel-space adversarial perturbations, and defenses against typographic/text overlay attacks. Pixel-space defenses include TRADES(Zhang et al., 2019), which regularizes the gap between predictions on clean vs. adversarial inputs, and VLM-adaptations like FARE that adversarially fine-tune CLIP's vision encoder; TGA-ZSR further aligns text-guided attention to stabilize zero-shot decisions under perturbations.(Schlarmann et al., 2024; Yu et al., 2024) Typographic defenses explicitly target overlaid text: PAINT(Ilharco et al., 2022) fine-tunes on a patch task and interpolates with the original weights to reduce typography-induced errors, while Defense-Prefix learns a single prefix token prepended to class names to make prompts resistant without changing CLIP's parameters;More recently, preference optimization for contrastive VLMs (Afzali et al., 2025) trains on pairwise preferences where the clean image–text pair is treated as preferred and its typographic-attack counterpart as dispreferred, steering the model to reject text-overlay errors while preserving clean accuracy. Despite improvements, these methods optimize proxy signals (logits, embeddings, attention, prefixes, or weight interpolation) rather than directly preserving cross-modal mutual information, leaving a gap our approach targets.

**Our standpoint: preserve information, not only surrogates.** We view cross-modal mutual information (MI) as a central quantity for VLM robustness. MI serves as the semantic link between image and text; when it is preserved, predictions remain more stable under perturbations because the shared content is intact. In contrast, proxy alignments can leave residual semantic drift under strong or diverse attacks. Motivated by this perspective, our objective centers training on the reduction of an MI gap: any attack-induced drop in MI is used as the learning signal to counteract the attack. Practically, we fine-tune the *vision* encoder while freezing the text encoder, encouraging adversarial embeddings to stay within the clean semantic clusters and to retain CLIP's zero-shot behavior.

## C  COST ANALYSIS IN DETAIL

**Common budget.** Across all variants we share a per–training-step budget consisting of: (i) one forward pass of the frozen encoder, $\phi_{Org}$ on the clean batch, (ii) a $k$-step inner maximization in pixel space, and (iii) one forward/backward pass of the trainable image encoder on the adversarial batch. The costs below describe only the *additional* work beyond this shared budget. All text label embeddings for ImageNet-1k are precomputed once and cached as a $512{\times}1000$ matrix (about 2.0 MB in fp32).

**FARE.** The loss is a squared $\ell_2$ distance between embeddings and therefore adds only head-level $O(Bd)$ arithmetic for batch size $B$ and embedding dimension $d{=}512$. No extra ViT passes and no extra trainable parameters.

**TeCoA.** The additional work is a matrix multiply between image embeddings ($B{\times}d$) and the cached text bank ($d{\times}C$ with $C{=}1000$), i.e., $O(BdC)$, plus cross-entropy. There are no extra ViT passes and no additional parameters.

**TGA-ZSR.** Attention maps are formed by correlating patch tokens with the (cached) class text vector. In our reproduce, this requires access to patch-level tokens for: (a) $\phi_{FT}$ on clean images and (b) $\phi_{FT}$ on adversarial images; the clean forward of $\phi_{Org}$ is already part of the shared budget. Each attention map induced by $\phi_{FT}$ entails a full ViT forward to obtain tokens ($B{\times}P{\times}512$ after projection with $P{=}49$ for $224^2$) and a patch–text correlation $O(BPd)$, followed by light normalization and distance computations. The dominant increment is thus two additional ViT passes of finetuned encoder, $\phi_{FT}$, and their token buffers.

**InfoGap (ours).** No extra ViT passes are introduced. All additional computation occurs on $(B{\times}512)$ embeddings: a small discriminator $D_\psi(x, y)$ on image–text pairs and the mutual-information terms (weighted and standard) computed with a non-parametric $T(x, y)$. The discriminator has $\approx 0.79\mathrm{M}$ parameters for the MLP head, which is less than $1\%$ of the CLIP ViT-B/32 vision tower which has 88M parameters. Consequently, the incremental wall-time and memory overhead are negligible.

**DPO / KTO (preference optimization).** In this implementation the *reference policy* is a frozen copy of `openai/clip-vit-base-patch32` created at initialization (all parameters have `requires_grad=False`). Each training step therefore adds one extra frozen forward of the current batch through the vision encoder to obtain reference image logits, followed by the usual matrix multiply with the cached text bank and the PO loss (log-ratios, sigmoids). No gradients flow through the reference and no additional trainable parameters are introduced.

**PPO.** The PPO objective consumes cached "old" log-probabilities from the policy and thus does not require any additional ViT forward within a step. Its extra computation reduces to elementwise probability ratios and clipping on $O(B)$ scalars, with no added parameters.

**Memory remarks.** The cached text bank occupies $\sim 2.0\,\mathrm{MB}$. For TGA-ZSR, the principal incremental memory is the storage of patch tokens of $\phi_{FT}$ for clean and adversarial batches (roughly $2 \times BPd$ floats for tokens alone; e.g., with $B{=}256$, $P{=}49$, $d{=}512$, $\approx 12.8\mathrm{M}$ ). InfoGap's discriminator activations on $(B{\times}512)$ are minor, and the MI terms are parameter-free. For DPO/KTO, keeping reference logits is $B{\times}C$ floats (e.g., $256{\times}1000 \approx 1\mathrm{M}$ floats or $\sim 4\,\mathrm{MB}$) if retained for bookkeeping.

# D  ON THE TIGHTNESS OF OUR BOUNDS

## D.1  BOUNDARY-RISK BOUND: THE ROLES OF $\tau_0$ AND $\epsilon$

The flip-risk bound of Theorem 3.1 involves a margin constant $\tau_0 > 0$ and the conditional MI terms $I(L; Z_{\mathrm{adv}} \mid Z)$ and $I(L; Z \mid Z_{\mathrm{adv}})$. The factor $\tau_0$ is inherited from standard margin/TV relaxations (Pinsker/Jensen steps) and is *not* optimized by our objective; any looseness via $\tau_0$ is thus structural to the inequality rather than method-specific.

We isolate the residual conditional MI

$$\epsilon \; := \; I(L; Z_{\mathrm{adv}} \mid Z),$$

and note that, in our setup, adversarial views are constructed by a label-agnostic transformation of the clean view.

**Lemma D.1** (Label-agnostic adversaries force $\epsilon = 0$). *If $Z_{\mathrm{adv}} = g(Z, \eta)$ for some (possibly stochastic) $g$ and noise $\eta$ independent of $L$ given $Z$, then $L \to Z \to Z_{\mathrm{adv}}$ forms a Markov chain and*

$$I(L; Z_{\mathrm{adv}} \mid Z) \; = \; 0.$$

*Proof.* By conditional independence, $p(L, Z_{\mathrm{adv}} \mid Z) = p(L \mid Z)\, p(Z_{\mathrm{adv}} \mid Z)$. Hence

$$I(L; Z_{\mathrm{adv}} \mid Z) \; = \; \mathbb{E}_Z \Big[ \mathrm{KL}\big(P_{L, Z_{\mathrm{adv}} \mid Z} \, \big\| \, P_{L \mid Z}\, P_{Z_{\mathrm{adv}} \mid Z}\big)\Big] \; = \; 0.$$

$\square$

**Corollary D.2** (Bound reduction under label-agnosticity). *Under the conditions of Lemma D.1, Corollary 3.2 reduces to*

$$R_{\mathrm{bdy}} \; \leq \; \frac{\sqrt{2}}{\tau_0} \, \sqrt{\Delta_{\mathrm{MI}}}, \qquad \Delta_{\mathrm{MI}} := I(V^+; U) - I(V^+; U').$$

*Remark* D.3 (Scope of $\epsilon$). The quantity $\epsilon$ is determined by whether the adversary uses label information; it is not directly controlled by our MI-gap objective. In the label-agnostic regime (our training setting) Lemma D.1 gives $\epsilon = 0$ exactly, and the flip-risk bound depends only on $\Delta_{\mathrm{MI}}$. If an alternative threat model were to inject label information, $\epsilon \geq 0$ could be nonzero; the general bound remains valid but $\epsilon$ then lies outside our optimization target.

## D.2 INFOGAP UPPER BOUND: WHERE SLACK ORIGINATES AND HOW THE DESIGN TARGETS IT

Recall Proposition 3.4 with $p$ and $q$ the clean and perturbed joints over $(U, V)$ and $w = \frac{p}{q}$:

$$R_{\text{bdy}} \leq \frac{\sqrt{2}}{\tau_0} \sqrt{I_p(U, V) - I_q(U, V)}$$

$$\leq \frac{\sqrt{2}}{\tau_0} \sqrt{\underbrace{I_q^{\text{IW}}(U, V) - I_q(U, V)}_{\text{information term}} + \underbrace{\text{KL}(p\|q)}_{\text{distribution term}}}$$

$$\leq \frac{\sqrt{2}}{\tau_0} \sqrt{\left(I_q^{\text{IW}}(U, V) - I_q(U, V)\right) + \chi^2(p\|q)}$$

$$\xrightarrow{\text{Surrogate Objective}} \mathcal{L}_{\text{outer}} = \left(\widehat{J}_q^{\text{IW}} - \widehat{J}_q\right) + \lambda \cdot \widehat{\chi}^2(p\|q) + \gamma \cdot \|u_{\text{adv}} - u\|_2^2$$

$$I_p(U, V) - I_q(U, V) \leq \underbrace{I_q^{\text{IW}}(U, V) - I_q(U, V)}_{\text{information term}} + \underbrace{\text{KL}(p\|q)}_{\text{distribution term}},$$

and this loss function stems from, when the text marginal is fixed ($p_V = q_V$), the identity

$$I_p = I_q^{\text{IW}} + \text{KL}(p\|q) - \text{KL}(p_U\|q_U) \tag{9}$$

exhibits a *structural* nonnegative residual $\text{KL}(p_U\|q_U)$.

**Observation D.4** (Sources of slack in the InfoGap upper bound). *The gap between the computable surrogate and the true MI gap admits the following nonnegative components:*

1. ***Structural slack*** $\text{KL}(p_U\|q_U)$ *from equation 9 (vanishes if $p_U = q_U$).*

2. ***Variational (critic) slack*** *from restricting the DV critic class in $I_q^{\text{IW}}$ and $I_q$.*

3. ***Density-ratio estimation slack*** *due to using an estimated $\hat{w}$ in place of $w = \frac{p}{q}$.*

4. ***Surrogate divergence slack*** *when replacing $\text{KL}(p\|q)$ by a tractable proxy (e.g., a $\chi^2$ penalty), using inequalities such as $\mathbb{E}_q[w \log w] \leq \mathbb{E}_q[(w-1)^2]$.*

*Remark* D.5 (How the regularizers address slack). By construction, our two regularizers directly *target* the slack terms in Observation D.4: (i) the $\chi^2$ anchor drives $w \to 1$, acting on the distribution term and reducing ratio–estimation error; (ii) the $\ell_2$ feature anchor discourages drift of $U$, promoting $p_U \simeq q_U$ and shrinking the structural slack $\text{KL}(p_U\|q_U)$. With a sufficiently expressive critic class, the variational slack contracts as well. These are design-level implications that follow from the definitions and inequalities above; they do not assert any particular empirical magnitude without measurement.

**Discriminator as a Qualitative Indicator of Reduced Slack.** Consider a logistic discriminator $D_\psi(u, v) \in (0, 1)$ trained with a standard binary cross-entropy loss to distinguish pairs drawn from the clean joint $p(u, v)$ versus the perturbed joint $q(u, v)$, using balanced sampling (equal proportions from $p$ and $q$). The Bayes-optimal predictor is

$$D^\star(u, v) = \frac{p(u, v)}{p(u, v) + q(u, v)},$$

so $D^\star = \frac{1}{2}$ if and only if $p = q$, which is equivalent to $w(u, v) = \frac{p}{q} = 1$. In that case $\chi^2(p\|q) = \mathbb{E}_q[(w-1)^2] = 0$ and, by Observation 3.6, the fixed-critic surrogate bias also vanishes. In Fig. 6(c), the *median* discriminator output drifts toward 0.5 over training. While this does not prove pointwise equality $p = q$, it is consistent with the two joints becoming harder to distinguish on average (i.e., $w \approx 1$), indicating smaller distribution/ratio–estimation slack; together with the $\ell_2$ anchor, this aligns with a reduction of the structural slack $\text{KL}(p_U\|q_U)$. Thus the plot serves as a qualitative sanity check that the objective is operating in the intended direction.

Table 8: Zero-shot clean (C) and robust (R) accuracies (%) for SigLIP ViT-B/32 under AutoAttack (AA) with $\epsilon = 2/255$. "Zeroshot Average" is the mean over the 13 zero-shot datasets. "C+R" is the sum of clean and robust zeroshot averages (reported on the R rows).

| Method | Accuracy | CIFAR10 | STL10 | CIFAR100 | Cars | Caltech101 | Pets | Flowers | DTD | EuroSAT | FGVC_Aircraft | PCam | ImageNet-R | Sketch | Zeroshot Average | C+R |
|---|---|---|---|---|---|---|---|---|---|---|---|---|---|---|---|---|
| TGA-ZSR (SigLIP) | C | 80.5 | 90.7 | 54.0 | 32.2 | 79.9 | 77.9 | 33.2 | 35.6 | 23.7 | 7.1 | 50.3 | 59.8 | 47.0 | 51.7 | |
| | R | 61.0 | 81.5 | 36.8 | 16.0 | 74.9 | 63.5 | 19.3 | 24.7 | 17.1 | 3.6 | 50.0 | 44.5 | 33.2 | 40.5 | 92.2 |
| FARE (SigLIP) | C | 77.0 | 92.0 | 58.3 | 61.5 | 81.1 | 83.0 | 51.6 | 48.6 | 16.5 | 8.7 | 50.0 | 65.7 | 50.3 | 57.3 | |
| | R | 55.1 | 81.3 | 38.9 | 29.7 | 70.1 | 63.4 | 28.0 | 29.3 | 12.9 | 2.3 | 48.0 | 46.1 | 35.2 | 41.6 | 98.8 |
| InfoGap (SigLIP, Ours) | C | 82.2 | 93.0 | 64.0 | 73.4 | 82.2 | 84.1 | 61.2 | 51.6 | 19.9 | 29.1 | 49.5 | 71.6 | 57.0 | 63.0 | |
| | R | 60.0 | 81.9 | 43.4 | 30.9 | 71.1 | 60.4 | 30.5 | 29.9 | 13.2 | 6.8 | 41.9 | 47.2 | 36.4 | 42.6 | 105.6 |

# E    ADDITIONAL EXPERIMENTS

## E.1    CROSS-ARCHITECTURAL GENERALIZATION

To verify that our approach is not tied to a specific backbone, we additionally fine-tune **SigLIP** using the same training recipe as our main CLIP experiments, except for swapping the vision encoder to SigLIP ViT-B/32 and increasing the outer-loop $L_2$ regularization coefficient between clean and adversarial embeddings to 20. SigLIP uses the same ViT-B/32 image backbone as CLIP but a deeper, higher-capacity text encoder, which provides richer text anchors and is particularly favorable to text-heavy defenses such as FARE.

Table 9 reports clean accuracy and robust performance under AutoAttack and PGD for both FARE and InfoGap in this SigLIP setting. Across all attack types and perturbation strengths, InfoGap achieves higher clean accuracy and consistently better clean-robust trade-offs . These results indicate that the MI-gap objective transfers to SigLIP without any architectural modification and that our conclusions are not specific to CLIP.

Table 8 reports zero-shot clean (C) and AutoAttack-robust (R) accuracies under $\epsilon = 2/255$ across the same set of zero-shot datasets.

Table 9: Average clean+robust score $(C + R)$ over 13 zero-shot datasets for SigLIP under different attacks.

| Method | AA$^2$ | AA$^4$ | PGD$^2$ | PGD$^4$ | CW | Avg |
|---|---|---|---|---|---|---|
| TGA-ZSR | 92.2 | 80.9 | 93.3 | 82.6 | 94.8 | 88.8 |
| FARE | 98.8 | 84.2 | 100.4 | 85.7 | 105.3 | 94.9 |
| InfoGap | **105.6** | **87.6** | **107.1** | **89.1** | **112.8** | **100.4** |

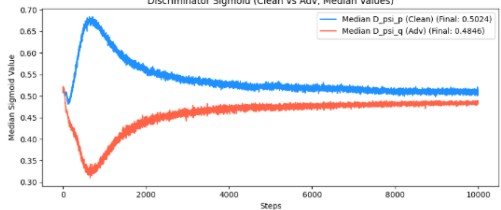

Figure 7: Training dynamics of the discriminator for the SigLIP experiment in Table 9.

## E.2    ADDITIONAL ABLATIONS AND ROBUSTNESS STUDIES

**Inner-loss ablations for InfoGap.**    We first compare three variants of InfoGap that differ only in the inner maximization objective while sharing the same outer MI-gap loss: (i) the baseline loss used in the main paper, (ii) an adaptive MI-based inner loss, and (iii) a label-aware cross-entropy (CE) inner loss. Table 10 reports the sum of clean and robust accuracies $(C+R)$ under AutoAttack (AA), PGD, and CW attacks at $\epsilon \in \{2/255, 4/255\}$. All three variants are stable; the CE inner loss yields the highest average score, indicating that making inner adversaries more label-informative slightly improves the final robustness while the MI-gap objective remains effective.

**Black-box robustness with Square Attack.**    We next evaluate robustness against the query-based black-box Square Attack (Andriushchenko et al., 2020), which perturbs inputs without accessing model parameters or gradients. Table 11 reports robust accuracy on 13 zero-shot datasets, to-

Table 10: Comparison of InfoGap training objectives. Each entry is the sum of clean and robust accuracy ($C+R$) across 13 zero-shot datasets under the specified attack and $\epsilon$. "Avg." is the average over all columns.

| Method | AA$^2$+Clean | AA$^4$+Clean | PGD$^2$+Clean | PGD$^4$+Clean | CW+Clean | Avg. |
|---|---|---|---|---|---|---|
| InfoGap (reported) | 83.9 | 69.4 | 85.6 | 71.8 | 88.6 | 79.9 |
| InfoGap (adaptive MI loss) | 84.4 | 68.4 | 86.5 | 69.9 | 92.9 | 80.4 |
| InfoGap (CE inner loss) | 84.5 | 71.4 | 86.3 | 72.9 | 89.1 | 80.8 |

gether with the clean zero-shot accuracy. InfoGap attains higher average robust accuracy than FARE (42.23% vs. 41.05%) while also achieving better clean performance (50.5% vs. 47.9%).

Table 11: Robust accuracy (%) under Square Attack (black-box). "Avg(Rob.)" is the mean robust accuracy across datasets; "Clean" is the clean zero-shot accuracy.

| Method | C10 | STL | C100 | Cars | Caltech | Pets | Flwrs | DTD | EuroSAT | Air | PCam | IN-R | Sktch | Avg(Rob.) | Clean |
|---|---|---|---|---|---|---|---|---|---|---|---|---|---|---|---|
| FARE | 60.0 | 86.1 | 35.8 | 23.3 | 74.7 | 66.9 | 22.9 | 25.5 | 13.6 | 6.6 | 50.2 | 40.6 | 27.5 | 41.05 | 47.9 |
| InfoGap | 69.5 | 88.0 | 42.4 | 21.1 | 72.2 | 64.7 | 27.0 | 29.1 | 13.9 | 4.5 | 49.2 | 41.6 | 25.8 | 42.23 | 50.5 |

**Additional pixel-wise baselines (PMG-AFT, Sim-CLIP).** To broaden the comparison, we further include PMG-AFT and Sim-CLIP, two recent robust fine-tuning methods for VLMs. Table 12 shows the $C+R$ scores under AA, PGD, and CW attacks. InfoGap achieves the best PGD$^2$+Clean score and matches or exceeds PMG-AFT and Sim-CLIP in most columns while using a single MI-based objective.

Table 12: Extended comparison with PMG-AFT and Sim-CLIP. Each entry is clean+robust accuracy ($C+R$) under the specified attack and $\epsilon \in \{2/255, 4/255\}$.

| Method | AA$^2$ | AA$^4$ | PGD$^2$ | PGD$^4$ | CW | Avg. |
|---|---|---|---|---|---|---|
| PMG-AFT | 79.6 | 70.5 | 81.1 | 72.6 | 83.5 | 77.5 |
| Sim-CLIP | 82.3 | 71.1 | 83.9 | 72.8 | 86.8 | 79.4 |
| InfoGap (reported) | 83.9 | 69.4 | 85.6 | 71.8 | 88.6 | 79.9 |

**Robustness at higher perturbation budgets.** Finally, we study robustness under a much stronger threat level, $\epsilon=8/255$, on the same pixel-wise benchmark used in the main paper. Table 13 reports clean accuracy as well as AA and PGD robust accuracy for FARE and InfoGap. All methods suffer large degradation at this extreme budget; InfoGap maintains higher clean accuracy but, as expected, the clean–robust trade-off is less favorable than in the small-$\epsilon$ regime where our main results are reported.

Table 13: Robustness under high-intensity attacks with $\epsilon = 8/255$. "Avg" is the mean over the 13 zero-shot datasets.

| Method | Setting | C10 | STL | C100 | Cars | Caltech | Pets | Flwrs | DTD | EuroSAT | Air | PCam | IN-R | Sktch | Avg |
|---|---|---|---|---|---|---|---|---|---|---|---|---|---|---|---|
| FARE | Clean | 73.3 | 90.0 | 49.6 | 34.5 | 78.7 | 78.0 | 31.4 | 30.8 | 15.9 | 10.0 | 50.2 | 48.6 | 31.7 | 47.9 |
| | AA8 | 5.9 | 18.7 | 5.4 | 0.3 | 23.8 | 2.3 | 1.4 | 5.5 | 2.7 | 0.0 | 47.6 | 5.3 | 3.9 | 9.4 |
| | PGD8 | 4.6 | 16.8 | 4.3 | 0.1 | 20.3 | 1.4 | 0.7 | 4.4 | 0.1 | 0.0 | 47.5 | 5.0 | 2.8 | 8.3 |
| InfoGap | Clean | 82.0 | 91.6 | 56.3 | 35.1 | 78.3 | 79.9 | 38.4 | 33.0 | 17.7 | 9.4 | 50.2 | 51.3 | 33.3 | 50.5 |
| | AA8 | 3.9 | 11.7 | 3.6 | 0.0 | 11.9 | 0.5 | 0.4 | 3.8 | 0.0 | 0.0 | 26.3 | 3.1 | 2.7 | 5.2 |
| | PGD8 | 2.9 | 9.5 | 2.9 | 0.0 | 9.3 | 0.0 | 0.1 | 2.6 | 0.0 | 0.0 | 25.1 | 2.6 | 2.0 | 4.4 |

# F    THE USE OF LARGE LANGUAGE MODELS.

All work presented in this paper is our own, with the exception of assistance for academic writing and grammatical review.

