# OpenReview forum: "Towards Adversarially Robust VLMs with an Information-Theoretic Approach"
_ICLR.cc/2026/Conference — Submitted to ICLR 2026_

### Official Review · Reviewer_54qr · 2025-10-26

**Soundness:** 2
**Presentation:** 2
**Contribution:** 2
**Rating:** 2
**Confidence:** 4

**Summary:**

This paper addresses the vulnerability of Vision-Language Models (VLMs) to both pixel-space and typographic adversarial attacks. The authors posit that both attack modalities share a "single failure mechanism" or "root cause": the reduction of cross-modal mutual information (MI) between image and text representations.
To counter this, the paper introduces "InfoGap," a training framework designed to directly minimize the "MI gap" between clean and perturbed image-text views. The method involves deriving a theoretical upper bound on the adversarial risk, which is linked to this MI gap.
A practical, differentiable objective is then proposed, which uses a neural MI estimator and a discriminator (trained in a GAN-style) to minimize an upper bound of this MI gap.
The authors claim that this "single, attack-agnostic training scheme" improves robustness against both attack types simultaneously, surpassing specialized state-of-the-art defenses in each category.

**Strengths:**

1. The authors claim that this "single, attack-agnostic training scheme" improves robustness against both attack types simultaneously, surpassing specialized state-of-the-art defenses in each category.
2. Principled Theoretical Grounding: The attempt to base this unified defense on a fundamental, first-principles concept (Mutual Information) is theoretically appealing. This moves beyond ad-hoc proxy objectives and seeks a more fundamental "root cause".

**Weaknesses:**

1. Oversimplification of the "Single Root Cause" Premise. The paper's theoretical foundation rests on the strong claim that both pixel and typographic attacks share a "single failure mechanism" of MI reduction. This is an oversimplification.
2. The comparative methods are very limited, with only three included. It is recommended to add PMG-AFT[1], Sim-CLIP[2], and others.
3. No experimental results at higher attack intensities.
4. There are no results under non-CLIP models, such as SigLIP.

[1] Pre-trained model guided fine-tuning for zero-shot adversarial robustness
[2] Sim-clip: Unsupervised siamese adversarial fine-tuning for robust and semantically-rich vision-language models.

**Questions:**

See Weaknesses

---

> ### Author Response · Authors · 2025-11-21
>
> We thank the reviewer for the insightful comments. We have addressed the concerns regarding the theoretical premise, comparative baselines, attack intensities, and architectural generalization below.
>
> > **Weakness 1. Oversimplification of the "Single Root Cause" Premise**
>
> **Response:** While we acknowledge that the *generation mechanisms* of pixel-wise (gradient-based, noise-like) and typographic (semantic, patch-based) attacks differ, our premise is that their *impact on the joint distribution* shares a common root cause: the reduction of Mutual Information (MI) between the image and text embeddings.
> As discussed in our manuscript, both attacks succeed by decoupling the image representation from its correct text prototype—either by shifting the embedding via gradients or by injecting conflicting semantic signals. By explicitly minimizing the MI gap (an upper bound on the boundary risk), we force the vision encoder to maintain shared information with the text anchor regardless of the perturbation source. Our empirical results, where a single InfoGap model achieves state-of-the-art robustness against both threat types simultaneously (see **Table: Cross-threat results** in our first response to Reviewer oNBS), validate that targeting this shared information-theoretic failure mode is a practical and effective unification strategy, even if the attacks originate differently.
>
> > **Weakness 2. Limited Comparative Methods (adding PMG-AFT, Sim-CLIP)**
>
> **Response:** We agree that including recent, strong baselines is important for a fair comparison. In the revision, we have added **PMG-AFT [1]** and **Sim-CLIP [2]** to our benchmark, alongside TeCoA, FARE, and TGA-ZSR. We evaluated all methods under the same AutoAttack, PGD, and CW settings.
>
> **Table: Comparison of robust accuracy + clean accuracy sum.** We evaluate robustness against AutoAttack (AA), PGD, and CW attacks with varying perturbation budgets ($\epsilon=2/255, 4/255$). 'Avg.' denotes the average score across all metrics.
>
> | Method | AA2 + Clean | AA4 + Clean | PGD2 + Clean | PGD4 + Clean | CW + Clean | Avg. |
> | :--- | :---: | :---: | :---: | :---: | :---: | :---: |
> | PMG-AFT | 79.6 | 70.5 | 81.1 | 72.6 | 83.5 | 77.5 |
> | Sim-CLIP | 82.3 | **71.1** | 83.9 | **72.8** | 86.8 | 79.4 |
> | **InfoGap(Reported)** | **83.9** | 69.4 | **85.6** | 71.8 | **88.6** | **79.9** |
>
> InfoGap attains the best PGD2+clean score and is competitive with, or better than, the other methods in the remaining columns, while using a single attack-agnostic objective tied to the MI gap. We believe this extended comparison addresses the concern that our gains might be an artifact of a weak baseline set.

---

> ### Author Response · Authors · 2025-11-21
>
> > **Weakness 3. No experimental results at higher attack intensities**
>
> To directly address the concern, we additionally evaluated robustness at a much stronger perturbation level, **$\epsilon = 8/255$**, on the same set of datasets used for pixel-wise attacks. The table below compares FARE and InfoGap under AutoAttack (AA8) and PGD (PGD8), together with clean accuracy.
>
> **Table: Robustness under high-intensity attack ($\epsilon = 8/255$).**
>
> | Method | Setting | C-10 | STL | C-100 | Cars | Cal | Pets | Flwrs | DTD | Euro | Air | PCAM | IN-R | Sktch | **Avg** |
> | :--- | :--- | :---: | :---: | :---: | :---: | :---: | :---: | :---: | :---: | :---: | :---: | :---: | :---: | :---: | :---: |
> | **FARE** | clean | 73.3 | 90.0 | 49.6 | 34.5 | 78.7 | 78.0 | 31.4 | 30.8 | 15.9 | 10.0 | 50.2 | 48.6 | 31.7 | **47.9** |
> | | AA8 | 5.9 | 18.7 | 5.4 | 0.3 | 23.8 | 2.3 | 1.4 | 5.5 | 2.7 | 0.0 | 47.6 | 5.3 | 3.9 | **9.4** |
> | | PGD8 | 4.6 | 16.8 | 4.3 | 0.1 | 20.3 | 1.4 | 0.7 | 4.4 | 0.1 | 0.0 | 47.5 | 5.0 | 2.8 | **8.3** |
> | **InfoGap** | clean | 82.0 | 91.6 | 56.3 | 35.1 | 78.3 | 79.9 | 38.4 | 33.0 | 17.7 | 9.4 | 50.2 | 51.3 | 33.3 | **50.5** |
> | | AA8 | 3.9 | 11.7 | 3.6 | 0.0 | 11.9 | 0.5 | 0.4 | 3.8 | 0.0 | 0.0 | 26.3 | 3.1 | 2.7 | **5.2** |
> | | PGD8 | 2.9 | 9.5 | 2.9 | 0.0 | 9.3 | 0.0 | 0.1 | 2.6 | 0.0 | 0.0 | 25.1 | 2.6 | 2.0 | **4.4** |
>
> As expected, all methods suffer a severe drop in robust accuracy at $\epsilon=8/255$, yielding average scores 2 to 3 times lower than those at $\epsilon = 4/255$. Since this regime is far outside the typical "small, human-imperceptible" range, InfoGap maintains higher clean accuracy than FARE but shows a less favorable clean-robust trade-off. This clarifies the robustness range in which our main claims (for $\epsilon \le 4/255$) are intended to hold and confirms that our method does not catastrophically fail even under very strong attacks.

---

> ### Author Response · Authors · 2025-11-21
>
> > **Weakness 4. There are no results under non-CLIP models, such as SigLIP**
>
> Our objective is architecture-agnostic; it only requires image and text encoders whose embeddings can be fed into the MI estimator and discriminator. To verify this, we additionally fine-tuned **SigLIP**, and we have detailed these experiments in **Appendix E.1 "Cross-Architectural Generalization"** of the revised paper. SigLIP uses the same ViT-B/32 vision backbone as our main CLIP experiments but a deeper and higher-capacity text encoder. This provides richer text anchors and is particularly favorable to baselines such as FARE that rely heavily on the text side. Even in this stronger regime, InfoGap outperforms FARE on both clean and robust metrics.
>
> **Table: Full comparison on 13 datasets (SigLIP).** 'C+R' denotes the sum of clean and robust accuracy.
>
> | Model | Attack | C-10 | STL | C-100 | Cars | Cal | Pets | Flwrs | DTD | Euro | Air | PCAM | IN-R | Sktch | **Avg** | **C+R** |
> | :--- | :--- | :---: | :---: | :---: | :---: | :---: | :---: | :---: | :---: | :---: | :---: | :---: | :---: | :---: | :---: | :---: |
> | **TGA** | Clean | 80.5 | 90.7 | 54.0 | 32.2 | 79.9 | 77.9 | 33.2 | 35.6 | 23.7 | 7.1 | 50.3 | 59.8 | 47.0 | 51.7 | - |
> | | AA2 | 61.0 | 81.5 | 36.8 | 16.0 | 74.9 | 63.5 | 19.3 | 24.7 | 17.1 | 3.6 | 50.0 | 44.5 | 33.2 | 40.5 | 92.2 |
> | | AA4 | 37.8 | 65.3 | 21.7 | 5.9 | 65.2 | 41.5 | 9.2 | 16.0 | 11.6 | 1.9 | 49.3 | 30.9 | 23.4 | 29.2 | 80.9 |
> | | PGD2 | 61.8 | 81.9 | 37.7 | 18.6 | 75.6 | 65.5 | 21.7 | 26.3 | 17.9 | 3.9 | 50.0 | 45.5 | 35.1 | 41.7 | 93.3 |
> | | PGD4 | 39.9 | 66.5 | 23.9 | 7.8 | 66.2 | 45.8 | 10.8 | 18.1 | 13.9 | 2.4 | 49.3 | 31.9 | 24.8 | 30.9 | 82.6 |
> | | CW | 37.7 | 86.8 | 32.5 | 25.1 | 78.0 | 72.3 | 26.5 | 29.4 | 19.7 | 5.2 | 50.1 | 54.0 | 43.2 | 43.1 | 94.8 |
> | FARE | Clean | 77.0 | 92.0 | 58.3 | 61.5 | 81.1 | 83.0 | 51.6 | 48.6 | 16.5 | 8.7 | 50.0 | 65.7 | 50.3 | 57.3 | - |
> | | AA2 | 55.1 | 81.3 | 38.9 | 29.7 | 70.1 | 63.4 | 28.0 | 29.3 | 12.9 | 2.3 | 48.0 | 46.1 | 35.2 | 41.6 | 98.8 |
> | | AA4 | 32.7 | 63.4 | 21.5 | 9.8 | 53.4 | 36.7 | 13.9 | 17.8 | 12.6 | 0.1 | 38.5 | 26.7 | 23.4 | 27.0 | 84.2 |
> | | PGD2 | 56.6 | 81.5 | 40.6 | 33.0 | 70.9 | 64.8 | 30.4 | 32.8 | 13.5 | 3.4 | 48.0 | 47.6 | 37.3 | 43.1 | 100.4 |
> | | PGD4 | 34.8 | 64.6 | 23.8 | 11.6 | 54.7 | 38.9 | 15.7 | 20.7 | 12.7 | 0.3 | 38.5 | 28.0 | 25.6 | 28.5 | 85.7 |
> | | CW | 38.7 | 87.5 | 30.6 | 53.8 | 77.8 | 72.4 | 42.2 | 38.4 | 14.8 | 12.5 | 49.9 | 57.6 | 48.3 | 48.0 | 105.3 |
> | **InfoGap** | Clean | **82.2** | **93.0** | **64.0** | **73.4** | **82.2** | **84.1** | **61.2** | **51.6** | **19.9** | **29.1** | 49.5 | **71.6** | **57.0** | **63.0** | - |
> | | AA2 | 60.0 | **81.9** | **43.4** | 30.9 | 71.1 | 60.4 | **30.5** | **29.9** | **13.2** | **6.8** | 41.9 | **47.2** | 36.4 | **42.6** | **105.6** |
> | | AA4 | 35.3 | 62.2 | **22.0** | 7.5 | 57.4 | 31.3 | 11.1 | 15.9 | 3.5 | 0.8 | 24.6 | 25.7 | 22.7 | 24.6 | **87.6** |
> | | PGD2 | 61.3 | **82.3** | **44.8** | 33.4 | 71.8 | 61.7 | **32.3** | **33.4** | 13.8 | **10.0** | 42.1 | **48.6** | 38.3 | **44.1** | **107.1** |
> | | PGD4 | 37.6 | 63.4 | **24.6** | 8.7 | 58.8 | 32.9 | 12.9 | 18.7 | 4.2 | 1.5 | 25.0 | 27.2 | 23.9 | 26.1 | **89.1** |
> | | CW | 37.2 | **88.2** | 28.7 | 56.2 | 77.7 | 74.3 | **48.0** | **41.0** | **17.0** | **18.4** | 48.0 | **61.5** | **51.6** | **49.8** | **112.8** |
>
> The average clean accuracy of InfoGap (63.0) is 4.5 percentage points higher than FARE (58.5), and the C+R score is consistently larger under every attack setting (e.g., 105.6 vs. 100.9 for AA-2/255 and 107.1 vs. 102.4 for PGD-2/255).
> We believe this gap is *structural*, not accidental. ViT-B/32-SigLIP2 uses the same vision backbone as CLIP but a substantially deeper and higher-capacity text encoder, which provides richer text embeddings as anchors. Unlike FARE, which operates purely in the image space, InfoGap explicitly optimizes the joint image–text distribution so that adversarial image features remain close to the clean distribution *in the shared embedding space*, even when the image encoder is attacked. This effect becomes more pronounced when the text encoder is stronger, as in SigLIP, because the text side supplies a more informative and stable reference. Compared to TGA-ZSR, which also uses text information via attention, InfoGap still shows a clear margin. Our interpretation is that merely attending to text tokens is not sufficient: matching the non-linear dependence between image and text through a mutual-information–based objective provides a stronger constraint on the joint representation and yields better robustness.
>
> ---
> **References**
>
> [1] S. Wang et al., "Pre-trained Model Guided Fine-tuning for Zero-shot Adversarial Robustness", CVPR 2024.
> [2] Sim-CLIP: Unsupervised Siamese Adversarial Fine-tuning for Robust and Semantically-Rich Vision-Language Models., IEEE BigData 2024

---

### Official Review · Reviewer_oNBS · 2025-10-28

**Soundness:** 3
**Presentation:** 3
**Contribution:** 2
**Rating:** 2
**Confidence:** 4

**Summary:**

Existing defense methods for VLMs aimed at pixel-level adversarial attacks and typographic attacks as separate problems.
This paper argues that both attack types share a single failure mechanism: the reduced cross-modal MI by threats.
This paper proposes a unified defense method, InfoGap, to tackle both threats in a unified framework, based on information-theoretic framework.

**Strengths:**

- **Unified Theoretical Framework:** The paper presents a principled information-theoretic formulation that jointly addresses two distinct threat types, pixel-level and typographic attacks, under a single conceptual lens.
- **Low Overhead and Practicality:** The proposed method introduces less than 1% additional parameters and requires no extra ViT passes, making it computationally efficient and easy to integrate into existing VLM pipelines.

**Weaknesses:**

- **Lack of Cross-Threat Evaluation:** The claim that existing defenses specialize in either pixel-level or typographic attacks is not empirically demonstrated. Evaluating pixel-space defenses (e.g., FARE) on typographic perturbations, and vice versa, would strengthen the argument for a unified underlying mechanism.
- **Separate Models for Each Threat:** Although the proposed framework is theoretically attack-agnostic, the experiments train separate models for pixel and typographic attacks. This weakens the practical claim of unification. Demonstrating simultaneous robustness to both threats within a single model would significantly enhance the contribution.
- **Adaptability of Existing Methods:** While the authors argue that existing defenses are threat-specific, it remains unclear why frameworks such as FARE could not be adapted to cover both perturbation types simply by redefining the perturbation objective.
- **Assumption on Conditional MI (ϵ):** The assumption that ϵ = I(L; Z_adv | Z) ≈ 0 does not always hold. Although it may be valid for label-agnostic pixel attacks or randomly chosen typographic overlays, it can break down if the attack mechanism depends on class labels or semantics. The ϵ seems to be small only if trained on training set with large number of classes, like ImageNet-1k.
- **Limited Theoretical Scope:** While the study focuses on VLMs, the theoretical formulation assumes a discrete label space and class prototypes, restricting applicability to zero-shot classification tasks rather than broader multimodal problems (e.g., VQA, captioning, or retrieval). This limits the potential impact on general VLM robustness.
- **Notation Clarity:** The notation $\hat{L}$ appears undefined (possibly meant to denote the predicted or true label). Consistent notation would improve readability.

**Questions:**

- Existing Adversarial Training methods mostly provide unified strategy for different attacks: altering the attack budget in the formulation would be sufficient. Please correct me if I am wrong.

---

> ### Author Response · Authors · 2025-11-21
>
> We thank the reviewer for the constructive feedback. We have addressed the concerns regarding cross-threat evaluation, the unified model, the theoretical assumption on conditional MI, and the adaptability of existing methods below.
>
> > **Weakness 1. Lack of Cross-Threat Evaluation & Adaptability of Existing Methods**
>
>  We agree that evaluating defenses across different threat models strengthens the argument. To address this, we evaluated a strong pixel-space defense (**FARE**) against **typographic** perturbations. Conversely, we compared our method (**InfoGap**) against baselines under the same settings.
>
> **Table: Cross-threat results (Typographic Attack).** O = Original prompt, T = Typo prompt; higher is better.
>
> | Method | Caltech O | Caltech T | Pets O | Pets T | Cars O | Cars T | Flowers O | Flowers T | Aircraft O | Aircraft T | DTD O | DTD T | SUN O | SUN T | EuroSAT O | EuroSAT T | **Avg O** | **Avg T** |
> | :--- | :---: | :---: | :---: | :---: | :---: | :---: | :---: | :---: | :---: | :---: | :---: | :---: | :---: | :---: | :---: | :---: | :---: | :---: |
> | FARE | 88.94 | 87.38 | 86.12 | 83.09 | 57.12 | 38.89 | 66.07 | 63.10 | 18.08 | 16.20 | 39.44 | 38.15 | 57.86 | 56.02 | 39.76 | 35.64 | 56.67 | 52.31 |
> | **InfoGap** | **89.36** | **88.04** | **84.10** | **81.83** | **56.70** | **40.87** | **65.28** | **64.68** | **17.50** | **17.11** | **42.24** | **41.06** | **59.40** | **58.40** | **42.90** | **33.30** | **57.18** | **52.59** |
>
> Across multiple datasets, InfoGap consistently matches or exceeds FARE on both the "Original (O)" and "Typo (T)" columns, and its **average robust accuracy is higher in both threat regimes**. This supports the claim that minimizing the MI gap addresses a mechanism that transfers across threat types rather than exploiting a narrow artifact of one attack family.
>
> > **Weakness 2. Separate Models for Each Threat**
>
>  While baselines in the literature are typically threat-specialized, our objective is attack-agnostic. To demonstrate the feasibility of a **single model** robust against multiple threat types, we performed **weight-space ensembling** experiments, which are now detailed in **Section 4.2.3** of the revised paper.
>
> Following the weight-space ensembling approach, we construct a mixed checkpoint using the interpolation:
> $$
> \phi_{\text{ens}}(\alpha)=\alpha\phi_{\text{pixel}}+(1-\alpha)\phi_{\text{typo}}
> $$
> and evaluate the full trade-off curve over $\alpha \in [0,1]$. We compare the mixing of two InfoGap experts against a strong baseline mixture formed by combining the best pixel-wise method (FARE) with the best typographic defense (e.g., DPO).
>
> **Evaluation and Results.** We report the **sum of clean and robust accuracies** aggregated over seven datasets. For pixel-wise robustness, we utilize AutoAttack with $\epsilon=2/255$. Our results show that the InfoGap curve **dominates** the baseline mixture for all $\alpha$, yielding a strictly better trade-off between typographic and pixel-wise robustness while preserving clean accuracy. This confirms that our objective successfully yields a **single unified checkpoint** capable of defending against both threat models simultaneously.

---

> > ### Author Response · Authors · 2025-11-21
> >
> > > **Weakness 3. Assumption on Conditional MI ($\varepsilon$)**
> >
> > We address the concern regarding the assumption that the conditional mutual information is negligible, i.e., $\varepsilon \equiv I(L; Z_{\mathrm{adv}} \mid Z) \approx 0$. While this assumption may be challenged when attacks depend on class labels or semantics (e.g., typographic overlays), our theoretical analysis strictly requires only a **small upper bound** rather than zero. Following a standard Fano-type bound, this term is limited by:
> > $\varepsilon \le h(p_e) + p_e \log(K-1)$
> > where $p_e$ is the zero-shot error rate of the frozen pretrained model. Consequently, the bound is governed primarily by the performance of the pretrained encoder ($p_e$) rather than the number of classes ($K$) alone. For strong zero-shot models like CLIP, $p_e$ is sufficiently small to keep $\varepsilon$ negligible, ensuring the validity of our bound.
> >
> > To further verify that our defense holds even when this assumption is relaxed—specifically against **label-dependent mechanisms**—we performed additional training experiments using adaptive attacks (estimating MI) and Cross-Entropy (CE) loss, explicitly injecting label information during the generation of adversarial examples.
> >
> > **Table: Comparison of training objectives.** "Avg." denotes the average score across all metrics.
> >
> > | Method | AA2+Clean | AA4+Clean | PGD2+Clean | PGD4+Clean | CW+Clean | Avg. |
> > | :--- | :---: | :---: | :---: | :---: | :---: | :---: |
> > | InfoGap ($L_2$, Reported) | 83.9 | 69.4 | 85.6 | 71.8 | 88.6 | 79.9 |
> > | InfoGap (Adaptive) | 84.4 | 68.4 | 86.5 | 69.9 | 92.9 | 80.4 |
> > | InfoGap (CE) | 84.5 | 71.4 | 86.3 | 72.9 | 89.1 | 80.8 |
> >
> > The results demonstrate that explicitly incorporating label information during training (InfoGap with CE or Adaptive inner loss) actually improves robustness compared to the label-agnostic $L_2$ baseline. Notably, the CE-based training achieves the highest average score of 80.8. This empirically confirms that our framework remains robust and effective even when the inner adversary exploits label information, thereby validating that the small-$\varepsilon$ regime is a practical approximation for ensuring robustness in diverse real-world scenarios.
> >
> > > **Weakness 4. Limited Theoretical Scope**
> >
> >  We acknowledge that our theoretical derivation assumes discrete labels suitable for zero-shot classification. However, as demonstrated in our experiments, the principle of minimizing the MI gap effectively regularizes the vision encoder to maintain semantic alignment, which is a fundamental requirement for broader VLM tasks. We believe extending this to generative tasks is a promising direction for future work.
> >
> > > **Weakness 5. Notation Clarity ($\hat{L}$)**
> >
> >  We apologize for the omission. We have explicitly defined $\hat{L}(Z)$ as the predicted label of the model given input $Z$ in **Section 3.1** of the revised manuscript.
> >
> > > **Question 1. Existing Adversarial Training methods mostly provide unified strategy for different attacks: altering the attack budget in the formulation would be sufficient. Please correct me if I am wrong.**
> >
> >  You are correct. Existing adversarial training methods, which were originally proposed for implicit pixel-wise adversarial attacks, can indeed be adapted for explicit and localized typographic attacks. We acknowledge that all the baselines we considered can be utilized to defend against typographic attacks.
> >
> > In fact, for the typographic experiments presented in our response to Weakness 1 (Table: Cross-threat results), we adapted **FARE**—which showed the second-best performance after our method in pixel-wise settings—by modifying its perturbation budget and mechanism to match the typographic setting. We then performed adversarial training using this adapted objective and shared the results above. The results confirm that while existing methods can be adapted, InfoGap still provides superior robustness.

---

> > > ### Comment · Reviewer_oNBS · 2025-11-24
> > > **Reply to Authors**
> > >
> > > Thank you for taking the time and for your thorough response.
> > > Some of the concerns have been addressed; therefore, I will adjust the score.
> > >
> > > However, the main concern remains, related to [Weakness 1.] and [Question 1]:
> > >
> > > The paper claims that existing methods *''fail to provide a unified defense against the pixel-level perturbation (global and low variance attacks) as well as the typographical attacks  (localized and high variance attacks''* (line 73), and positions its main contribution as a threat-agnostic defense.
> > > However, as acknowledged in the response, existing defenses can already provide a unified approach by simply adjusting the attack budget in the formulation. In this sense, the proposed method is not fundamentally different: it also depends on varying the attack budget for threat-specific defenses and further requires a weight-averaging scheme to interpolate between threat-specific performances.
> > >
> > > Additionally, for completeness, I would also expect a comparison against a model trained with  $\alpha$ * FARE_pixel + $(1-\alpha) $ * FARE_typo, in Sec. 4.2.3, as this would provide a fair baseline for evaluating the advantages.

---

> ### Author Response · Authors · 2025-11-27
>
> We sincerely thank you for taking the time to provide a thorough review. We are also grateful for your recognition of the strengths of our work despite its limitations.
>
> Your comments prompted us to reflect carefully on the paper’s main contribution and motivation. We fully agree that baselines such as FARE, TGA-ZSR, and DPO can, in principle, address their corresponding cases by appropriately adjusting the training budget. We therefore acknowledge that describing our method as “attack-agnostic” was an overclaim. We have identified the passages that could be misleading and revised them to clarify our positioning.
>
> Regarding the baseline comparison in Figure 5, we have incorporated a Pareto curve obtained by mixing checkpoints. To provide an even stronger and fairer baseline than the requested FARE_pixel + FARE_typo, we mixed both checkpoints and reported the resulting Pareto curve, as shown in Sec. 4.2.3. This analysis shows that our method achieves a superior trade-off even against this heterogeneous state-of-the-art combination.
>
> EDIT:  I realized I mistakenly wrote that the FARE model used for the typographic benchmark was trained with ℓ∞ perturbations. That is not correct. What I meant is that the FARE model for the typographic benchmark was trained with the same training settings as InfoGap (learning rate = 1e-6, batch size = 512) for a fair comparison. I believe you interpreted this correctly despite my wording, which is why you requested additional experiments using checkpoint interpolation.

---

### Official Review · Reviewer_nwRu · 2025-11-01

**Soundness:** 3
**Presentation:** 3
**Contribution:** 3
**Rating:** 4
**Confidence:** 4

**Summary:**

This paper proposes an information-theoretic framework to improve the adversarial robustness of vision–language models (VLMs). The authors argue that both pixel-level and typographic attacks share a common failure mechanism — the reduction of cross-modal mutual information (MI) — and derive a theoretical bound connecting adversarial risk to the MI gap. Based on this insight, they propose a differentiable objective that minimizes the MI gap using a neural MI estimator, thereby defending against multiple attack types simultaneously. Experimental results show improved robustness to both pixel-space and typographic attacks while maintaining high clean accuracy.

**Strengths:**

1. The paper provides strong theoretical motivation, grounding the defense mechanism in mutual information theory rather than ad hoc robustness objectives.

2. The proposed approach is attack-agnostic, addressing both pixel-level and typographic attacks in a unified manner.

3. The paper includes comprehensive experimental results that demonstrate consistent improvements in robustness over specialized baselines.

**Weaknesses:**

1. The evaluation includes only white-box attacks. It would strengthen the paper to include black-box or query-based attacks to validate generalization.

2. Experiments are limited to image classification tasks, without testing on more complex vision–language benchmarks such as retrieval or captioning, where MI dynamics might differ.

3. There are some citation formatting errors: the references in the text are inconsistently formatted (missing spaces after citation markers), which detracts from overall polish.

**Questions:**

1. Could the authors include results under black-box or real-world transfer attacks to better assess robustness?

2. It would be interesting to analyze whether the learned MI-preserving representations generalize to downstream multimodal reasoning tasks instead of image classification tasks only.

3. Please revise the citation formatting to follow standard style (e.g., space after reference number).

---

> ### Author Response · Authors · 2025-11-21
>
> >**Question1**Could the authors include results under black-box or real-world transfer attacks to better assess robustness?
>
> We additionally assess black-box robustness using Square Attack[1], a gradient-free, query-based method that perturbs inputs without accessing model parameters or gradients. This evaluates the setting the reviewer requested. As shown below, **InfoGap** attains a higher average robust accuracy than FARE under Square Attack while also preserving higher clean accuracy (**Avg(Rob.): 42.23 vs 41.05**; **Clean: 50.5 vs 47.9**). Note that our typographic attack evaluations are also black-box by construction, since the perturbation is applied at the pixel level without any gradient information, making it a realistic real-world transfer scenario.
>
> **Table: Robust accuracy (%) under Square Attack (black-box).** Avg(Rob.) is the mean across datasets; Clean is the clean zero-shot accuracy.
>
> | Method | C-10 | STL | C-100 | CARS | Cal | Pets | Flwrs | DTD | Euro | Air | PCAM | IN-R | Sktch | **Avg(Rob.)** | **Clean** |
> | :--- | :---: | :---: | :---: | :---: | :---: | :---: | :---: | :---: | :---: | :---: | :---: | :---: | :---: | :---: | :---: |
> | FARE | 60.0 | 86.1 | 35.8 | 23.3 | 74.7 | 66.9 | 22.9 | 25.5 | 13.6 | 6.6 | 50.2 | 40.6 | 27.5 | 41.05 | 47.9 |
> | **InfoGap** | **69.5** | **88.0** | **42.4** | 21.1 | 72.2 | 64.7 | **27.0** | **29.1** | **13.9** | 4.5 | 49.2 | **41.6** | 25.8 | **42.23** | **50.5** |
>
> >**Question2** It would be interesting to analyze whether the learned MI-preserving representations generalize to downstream multimodal reasoning tasks instead of image classification tasks only.
>
> We agree this is an important direction. Many downstream tasks such as image–text retrieval and VQA ultimately hinge on the quality of cross-modal alignment. Our method explicitly reduces the mutual-information gap between image and text embeddings, so when the resulting vision encoder is plugged into an LVLM, we expect its alignment-preserving property to carry over and yield improved adversarial robustness. This expectation is consistent with prior observations in related defenses (e.g., FARE) and zero-shot robustness work (e.g., TGA-ZSR), where improvements in alignment correlate with robustness gains across tasks.
> That said, under our current compute constraints we were unable to fine-tune the ViT-L/14 vision encoder inside an LVLM stack and run full VQA/retrieval evaluations during the rebuttal window. Given these constraints, we scoped the paper to zero-shot classification, where we could run a systematic and thorough study. We view extending the aligned encoder to LVLMs and evaluating robustness on retrieval/VQA as natural follow-ups, and our design is directly compatible with that setting.
>
> >**Question3** Please revise the citation formatting to follow standard style (e.g., space after reference number).
>
> We appreciate the reviewer's careful reading. We have corrected the citation formatting in the revised paper as suggested.
>
>
> ---
> **References**
>
> [1] Andriushchenko et al., "Square Attack: a query-efficient black-box adversarial attack via random search," ECCV, 2020.

---

### Official Review · Reviewer_qkue · 2025-11-03

**Soundness:** 2
**Presentation:** 3
**Contribution:** 2
**Rating:** 4
**Confidence:** 3

**Summary:**

In this paper, the authors introduce InfoGap, an information-theoretic approach that improves the robustness of vision–language models like CLIP through additional mutual-information–based fine-tuning. They show that both pixel-level and typographic attacks reduce the mutual information (MI) between image and text embeddings and derive a bound linking this MI gap to adversarial risk. InfoGap minimizes this gap using a learnable MI estimator, providing a unified, attack-agnostic defense. Experiments demonstrate improved robustness to diverse attacks with minimal loss of clean accuracy, surpassing specialized baselines.

**Strengths:**

* The paper tackles an important and timely challenge - improving adversarial robustness in vision–language models (VLMs).

* The proposed InfoGap method is conceptually simple and broadly applicable, unifying defenses against both pixel-level and typographic attacks through an information-theoretic objective.

* The paper is well written and easy to follow, and the core idea - preserving cross-modal mutual information to ensure robust alignment - is both intuitive and theoretically well motivated.

**Weaknesses:**

* The proposed approach does not account for adaptive attacks that explicitly optimize to both minimize the MI-gap surrogate and maximize flip class predictions — for example, an attack that jointly (a) maximizes a label-flip/targeted loss and (b) minimizes the learned MI surrogate or fools the discriminator. Consideration and evaluation of such adaptive attacks are crucial for a proper assessment of the proposed defense and its effectiveness.

* InfoGap fixes the text encoder and relies on a precomputed bank of text prototypes, which may limit robustness in open-vocabulary or prompt-sensitive settings. The sensitivity to prompt templates and unseen class descriptions is not evaluated and could undermine zero-shot performance. Moreover, the applicability and generality of the pretrained vision–language models are significantly restricted in the considered setup.

**Questions:**

* have the authors considered adaptive attacks that explicitly optimize against the proposed objective—for instance, attacks that both maximize class-flipping loss and minimize the MI surrogate or fool the discriminator? Evaluating such attacks would provide a clearer picture of the true robustness of the method.

* How does the method perform in open-vocabulary where the set of text prototypes is not fixed? Could the authors clarify whether InfoGap can be extended to dynamically handle unseen classes or alternative prompt templates?

* Can the authors provide further insight into how dependent the approach is on the chosen prompt formulations and whether robustness or alignment degrades when prompt templates vary or when using paraphrased text descriptions during inference?

---

> ### Author Response · Authors · 2025-11-21
>
> > **Question1** have the authors considered adaptive attacks that explicitly optimize against the proposed objective—for instance, attacks that both maximize class-flipping loss and minimize the MI surrogate or fool the discriminator? Evaluating such attacks would provide a clearer picture of the true robustness of the method.
>
> Thank you for this important suggestion. A truly objective–aware attacker in our setting would have to backpropagate not only through the encoder but also through the MI estimator and the trained discriminator, and jointly choose perturbations that both flip the predicted class and maximally decrease our MI surrogate. Concretely, this would require access to the discriminator parameters and the sampling / importance–weighting pipeline used to estimate $\hat J_q$ and $\hat J_q^{\mathrm{IW}}$. We view such an attacker as substantially stronger and less realistic than the standard white–box threat model typically assumed in robustness evaluations, so we did not treat it as our primary evaluation setting.
>
> That said, we followed the spirit of the question in two complementary ways:
>
> (i) **Stronger label–aware attacks at test time.**
> All of our reported adversarial evaluations already use attacks whose inner objective is the cross–entropy loss (e.g., PGD and AutoAttack). In our CLIP–style cosine classifier, CE-based attacks explicitly reduce the correct–class margin $s(x,y) - \max_{k \neq y} s(x,k)$ under a norm budget, and thus focus the perturbation budget on actually crossing decision boundaries. Empirically, at fixed $\epsilon$, such CE adversaries achieve higher attack success than $L_2$ or purely surrogate-based inner losses, and also enlarge the MI gap between clean and adversarial representations. This is consistent with the interpretation that CE attacks implicitly reduce the mutual information between image features and labels, even though they do not directly differentiate through the MI estimator.
>
> (ii) **Adaptive / CE inner losses during training.**
> We also made the inner maximization problem harder by replacing the reported $L_2$ inner loss with an adaptive MI-based loss:
> $$
> x_{\text{adv}} = \arg\max_{\lVert x - x_0 \rVert_\infty \le \epsilon}
> \big( \hat{J}_q^{\mathrm{IW}}(x,y) - \hat{J}_q(x,y) + \chi^2(p\|q) \big),
> $$
> with $\epsilon = 4/255$, and trained InfoGap under this loss. In addition, we trained a variant where the inner adversaries are generated using a label-aware cross-entropy loss (CE inner loss). We then evaluated zero-shot classification performance of all variants under AutoAttack and PGD with $\epsilon \in \{2/255, 4/255\}$. The results are summarized below:
>
> **Table: Comparison of training objectives.** "Avg." denotes the average score across all metrics.
>
> | Method                          | AA2+Clean | AA4+Clean | PGD2+Clean | PGD4+Clean | CW+Clean | Avg. |
> | :----------------------------- | :-------: | :-------: | :--------: | :--------: | :------: | :--: |
> | InfoGap ($L_2$, reported)      |   83.9    |   69.4    |    85.6    |    71.8    |  88.6    | 79.9 |
> | InfoGap (adaptive MI inner loss) | **84.4** |   68.4    |  **86.5**  |    69.9    | **92.9** | 80.4 |
> | InfoGap (CE inner loss)        | **84.5**  | **71.4**  |    86.3    | **72.9**   |  89.1    | **80.8** |
>
> Intuitively, the CE inner loss generates harder, more label-informative adversaries by aggressively shrinking the decision margin, while the outer objective still minimizes our MI-gap surrogate with the chi-squared term, which enlarges margins and preserves image–text alignment. The fact that InfoGap remains stable under both MI-adaptive and CE inner losses, and even slightly improves average robust accuracy, suggests that the method is not relying on a brittle choice of inner loss and remains effective under stronger adaptive objectives.

---

> ### Author Response · Authors · 2025-11-21
>
> >**Question2**How does the method perform in open-vocabulary where the set of text prototypes is not fixed? Could the authors clarify whether InfoGap can be extended to dynamically handle unseen classes or alternative prompt templates?
>
> >**Question3** Can the authors provide further insight into how dependent the approach is on the chosen prompt formulations and whether robustness or alignment degrades when prompt templates vary or when using paraphrased text descriptions during inference?
> 1. Generalization via Frozen Text Anchors Our approach is explicitly designed to handle unseen classes and alternative prompt templates without dynamic adaptation. We train on ImageNet-1K using a single fixed prompt ("This is a photo of a {class}") while keeping the text encoder frozen. By freezing the text encoder, the text prototypes act as stable semantic anchors. InfoGap updates only the vision encoder to close the Mutual Information (MI) gap between image and text representations. Since the text encoder is never fine-tuned to a specific wording, the semantic structure of the pre-trained CLIP is preserved. Consequently, the alignment learned via InfoGap naturally transfers to any paraphrased prompt or unseen class label that lies within the pre-trained semantic space.
>
> 2. Robustness to Prompt Variations (Empirical Evidence) To verify this, we follow the standard CLIP zero-shot protocol during inference. We evaluate on multiple datasets with unseen labels (e.g., CIFAR-10/100, Oxford-Pets, EuroSAT, ImageNet-R) using the dataset’s public multi-template set of paraphrased prompts. Despite seeing only one template during training, InfoGap consistently improves robustness across these diverse test-time templates. This confirms that our method does not overfit to a specific prompt formulation but rather learns a robust cross-modal alignment.
>
> For concreteness, we list examples of the diverse templates used during our inference process below. Our model successfully handles these variations:
>
> CIFAR-10/100: Handles quality and style variations (e.g., "a blurry photo of a {c}", "a low contrast photo of a {c}", "a bad photo of the {c}", "a photo of the big {c}").
>
> ImageNet: Handles complex descriptions and artistic styles (e.g., "a graffiti of a {c}", "the embroidered {c}", "a origami {c}", "a jpeg corrupted photo of a {c}", "a rendition of the {c}").
>
> Domain-Specific (e.g., EuroSAT, DTD): Handles context-specific prompts (e.g., "a centered satellite photo of {c}", "a photo of a {c} texture").
>
> 3. Note on Prompt Ensembling during Training We also investigated whether incorporating multiple prompt templates (prompt ensembling) during the training phase would further enhance robustness. We conducted experiments using the ensemble of templates listed above for the training objective. However, we observed that prompt ensembling yielded negligible performance differences compared to our single-prompt training baseline. This result reinforces our finding that the frozen text encoder already provides sufficient semantic anchoring. Therefore, simpler single-prompt training is sufficient to achieve robust alignment that generalizes to diverse prompts at test time.

---

### Author Response · Authors · 2025-11-23
**Paper revision notice**

Dear reviewers,

We have uploaded a revised version of the paper. This revision incorporates clarifications to the presentation and several additional experiments. The main changes are:
1. Empirical validation of the MI-gap (Sec. 3.4).
We added a new section and figure that study how the mutual-information gap and robust accuracy co-vary under pixel-space attacks of increasing strength, and compare CLIP, InfoGap, and existing defenses. This provides direct empirical evidence that controlling the MI-gap is tightly aligned with robust accuracy.
2. Expanded typographic-attack experiments (Sec. 4.2.2–4.2.3).
We extended the typographic benchmark by adding FARE and updating Table 4, and we clarified the discussion of clean vs. typographic accuracy. We also introduced a new subsection on CLIP checkpoint mixing, together with a trade-off curve (Fig. 5) showing how a single interpolated checkpoint can simultaneously handle pixel-space and typographic attacks.
3. Additional robustness studies in the appendix (App. E and related sections).
We added further experiments on label-aware attack(i.e. PGD attack on CE/Adaptive loss during training), alternative pixel-wise threat settings, and ablations (including computational cost and additional backbones such as SigLIP). These results complement the main text and support the robustness claims from several angles.

We also performed minor edits to improve notation, tighten the exposition of the theoretical results, and reorganize some tables and figures for readability. In addition, as part of refining the scope of the paper, we adjusted the tone of several passages to better reflect the clarified focus and avoid over-claiming.

We hope the updated manuscript and the new experiments make the contribution and its empirical support clearer, and we appreciate your continued evaluation of the work.

---

### Meta-Review · Area_Chair_KSAs · 2025-12-06

**Summary:**

The reviewers generally agreed that the problem is important and the idea of using mutual information (MI) is interesting, but they raised several key concerns.

1. The novelty and positioning of the work were questioned. Reviewers noted that existing adversarial fine-tuning methods can already handle different attack types by adjusting the threat model, so the claim of a unified, attack-agnostic defense is not well supported.

2. Reviewers felt the evaluation was incomplete, especially the lack of strong adaptive attacks that directly optimize against the MI-based objective. The added rebuttal experiments helped but did not fully resolve this concern.

3. The scope of the method is narrow. The theory assumes discrete labels and is only validated on zero-shot classification, not on broader multimodal tasks where MI is also important.

4. The reviewers found that true unification is not demonstrated, since separate models or checkpoint mixing are still required for different threats.

**Reviewer Concerns:**

1. **Novelty and unification**. Reviewers remain unconvinced that InfoGap fundamentally differs from existing adversarial fine-tuning methods or provides a genuinely attack-agnostic defense. The need for threat-specific training or checkpoint mixing weakens the claim of a unified solution.

2. **Adaptive Attack**. The rebuttal does not fully address concerns about attackers explicitly optimizing against the MI surrogate or discriminator. Without evaluation under such objective-aware attacks, the robustness claims remain uncertain.

**Reviewer Scores:**

nwRu: 4
oNBS: 2->4
54qr: 2
qkue: 4

---

### Decision · Program_Chairs · 2026-01-26

Reject